# WATERMARKING USING SEMANTIC-AWARE SPECULATIVE SAMPLING: FROM THEORY TO PRACTICE

## ABSTRACT

Statistical watermarking offers a theoretically-sound method for distinguishing machine-generated texts. In this work, we first present a systematic theoretical analysis of the statistical limits of watermarking, by framing it as a hypothesis testing problem. We derive nearly matching upper and lower bounds for (i) the optimal Type II error under a fixed Type I error, and (ii) the minimum number of tokens required to watermark the output. Our rate of $\Theta(h^{-1}\log(1/h))$ for the minimum number of required tokens, where $h$ is the average entropy per token, reveals a significant gap between the statistical limit and the $O(h^{-2})$ rate achieved in prior works. To our knowledge, this is the first comprehensive statistical analysis of the watermarking problem. Building on our theory, we develop **SEAL** (**S**emantic-awar**E** specul**A**tive samp**L**ing), a novel watermarking algorithm for practical applications. SEAL introduces two key techniques: (i) designing semantic-aware random seeds by leveraging a proposal language model, and (ii) constructing a maximal coupling between the random seed and the next token through speculative sampling. Experiments on open-source benchmarks demonstrate that our watermarking scheme delivers superior efficiency and tamper resistance, particularly in the face of paraphrase attacks.

| Results | | Best achievable Type II error | Number of required tokens |
|---|---|---|---|
| Upper bounds | Ours | $(1-\alpha)^{1/\kappa}$ | $h^{-1} \cdot \log(1/h)$ |
| | Previous | N/A | $h^{-2}$ |
| Lower bounds | Ours | $(1-\alpha)^{1/\kappa}$ | $h^{-1} \cdot \log(1/h)$ |
| | Previous | N/A | N/A |

(a) Theoretical results.

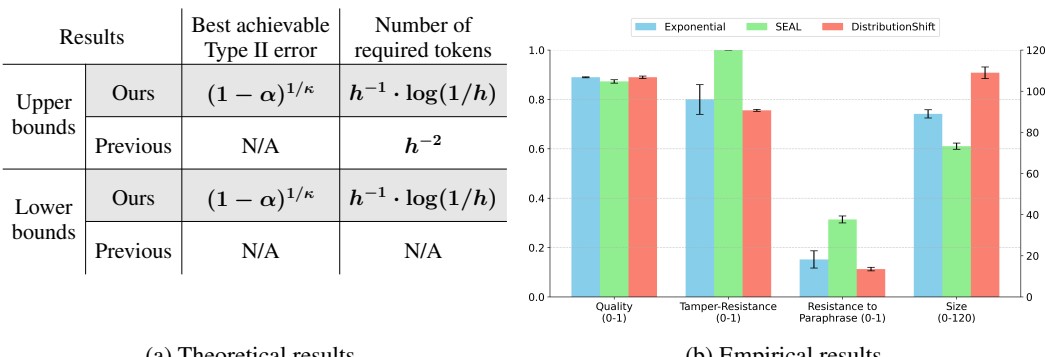

(b) Empirical results.

Figure 1: **Overview of our results.** In (a), we present upper and lower bounds for the best achievable Type II error and the number of required tokens, where $\alpha$ is the Type I error, $\kappa$ is the probability of the most likely output, and $h$ is the entropy per token. Our theoretical bounds demonstrate near-optimality, providing significant improvements over existing results. In (b), we empirically compare SEAL with two of the state-of-the-art watermarking methods, the exponential scheme (Aaronson, 2022b) and the distribution shift scheme (Kirchenbauer et al., 2023a), at sampling temperature 1. SEAL maintains comparable quality while being more tamper-resistant to various perturbations and efficient in size (smaller size indicates requiring fewer tokens to watermark).

## 1 INTRODUCTION

Large Language Models (LLMs) have revolutionized natural language tasks (Brown et al., 2020; Bubeck et al., 2023; Chowdhery et al., 2023). LLMs excel at generating human-like texts that can be difficult to distinguish from those written by human (OpenAI, 2023; Team et al., 2023; Anthropic,

2024). This capability raises several societal concerns regarding the misuse of LLM outputs. For instance, LLM-generated texts might contaminate training datasets for future language models (Shumailov et al., 2023; Das et al., 2024), facilitate the spread of misleading information (Zellers et al., 2019; Vincent, 2022), or be used in academic misconduct (Jarrah et al., 2023; Milano et al., 2023). The widespread use of LLMs underscores the need for effective detection methods to identify whether a human-like text is produced by an LLM system.

To detect machine-generated content, recent research works (Kirchenbauer et al., 2023a; Kuditipudi et al., 2023; Christ et al., 2023; Yoo et al., 2023; Fernandez et al., 2023; Fu et al., 2023; Wang et al., 2023; Yang et al., 2023; Liu et al., 2023; Zhao et al., 2023; Hu et al., 2023; Koike et al., 2023; Li et al., 2024; Ren et al., 2024; Liu & Bu, 2024; Hou et al., 2024b;a) have proposed the use of *statistical watermarks*. These are signals embedded within generated texts to reveal their source. Statistical watermarking modifies the decoding mechanism of LLMs so that the text output $X$ is sampled *jointly* with a sequence of random seeds $S$. Consequently, outputs from a watermarked LLM are always correlated with the accompanying random seeds, while texts generated from other sources (e.g., human writers) are not. Therefore, detecting whether $X$ is generated by an LLM reduces to testing the independence of $S$ and $X$. This procedure can be framed as a hypothesis test with two competing hypotheses:

$$\mathbf{H}_0 : X \text{ is sampled independently from } S,$$

$$\mathbf{H}_1 : (X, S) \text{ is sampled from a joint distribution}$$

Here, the null hypothesis $\mathbf{H}_0$ implies that $X$ is not generated by the watermarked LLM, while the alternative hypothesis $\mathbf{H}_1$ implies that $X$ is sampled from the watermarked LLM with joint distribution $\mathcal{P}$.

The major benefit of statistical watermarking is that it comes with formal statistical guarantees (Aaronson, 2022b; Kuditipudi et al., 2023; Christ et al., 2023; Zhao et al., 2023; Li et al., 2024). Specifically, these guarantees control the following three quantities:

1. Distortion (Bias): The distance between the watermarked LLM and the original LLM.
2. Type I error: The probability that an independently sampled output $X$ is incorrectly rejected as being generated by the watermarked LLM.
3. Type II error: The probability that an output from the watermarked LLM fails to be detected.

Despite these statistical guarantees, theoretical understanding of their statistical limits remains unsatisfied. For example, one may wonder, *à la* Neyman-Pearson:

*Q1: What is the best achievable Type II error under a fixed Type I error constraint?*

In practice, users often modify text outputs from large language models (LLMs) or insert their own human-written content. This complicates the task of identifying whether an article is entirely generated by LLMs, while highlighting the need to detect specific sequences of words produced by LLMs. For practitioners, a key consideration is to minimize the number of tokens used for watermarking and detection of a sequence. This leads to an important question:

*Q2: What is the minimum number of tokens required to watermark the output?*

To the best of our knowledge, these questions have not been addressed in prior research. As discussed in Section 1.1, previous rates on the number of tokens consistently fails to surpass $h^{-2}$ where $h$ is the average entropy per token, and it remains unknown whether this rate is optimal at all.

A key objective of theoretical analysis is to guide the development of practical algorithms. In light of the aforementioned questions raised, it is essential to translate theoretical insights into improved watermarking design. Therefore, we pose a final question:

*Q3: Can statistical theory be leveraged to design more effective and robust watermarking algorithms?*

In this paper, we address these three questions. Our contributions can be summarized in three-fold:

1. First, we establish the optimal Type II error achievable subject to a Type I error upper bound of $\alpha$. We distinguish between two scenarios: in the model-nonagnostic case, where

the detector has access to the fixed generation model $\rho$, the instance-dependent optimal Type II error is given by $\sum_{x \in \Omega : \rho(x) > \alpha} (\rho(x) - \alpha)$. In the model-agnostic case, where the generation model belongs to a known class $\mathcal{H}\kappa = \{\rho : \max_{\omega \in \Omega} \rho(\omega) \leq \kappa\}$ but is itself inaccessible to the detector, the minimax-optimal Type II error scales as $(1 - \alpha)^{1/\kappa}$. To our knowledge, this is the first formal result on the statistical limits of watermarking.

2. Second, we derive the minimum number of tokens required to watermark LLM outputs. In the theoretical setting where tokens are independent and identically distributed (iid) with entropy $h$, we demonstrate (nearly) matching upper and lower bounds on the number of required tokens, with the rate $\frac{\log(1/h)}{h}$ explicitly dependent on the entropy. Generalizing to non-iid tokens, which are practical, we show that if $n$ and $H$ satisfy $n, H \gtrsim \log \frac{1}{\alpha} + \log \log \frac{1}{\beta}$, then outputs with entropy at least $H$ and length at least $n$ can be watermarked with Type I error $\leq \alpha$ and Type II error $\leq \beta$. Our results improve upon the previous bound of $h^{-2}$, revealing a gap between existing algorithms and theoretical limits.

3. Finally, drawing on the theory we develop, we propose a novel watermarking algorithm, **S**emantic-awar**E** specul**A**tive samp**L**ing (**SEAL**), that bridges the gap between the theoretically-optimal watermark and state-of-the-art practical implementations. Our approach overcomes the limitations of prior methods by allowing the random seeds to adapt to the *semantic information* of preceding tokens, and furthermore enhances statistical efficiency through speculative sampling between random seed and token distributions. Experiments on MarkMyWords benchmark (Piet et al., 2023) show that our watermarking method achieves superior efficiency and tamper resistance, and especially outperforms state-of-the-art approaches in tamper resistance to paraphrase attacks.

## 1.1 RELATED WORKS

Watermarking is a powerful white-box method for detecting LLM-generated texts (Tang et al., 2023). Watermarks can be injected either into a pre-existing text (edit-based watermarks) or during the text generation (generative watermarks), while our work falls in the latter category. Edit-based watermarking (Rizzo et al., 2019; Abdelnabi & Fritz, 2021; Yang et al., 2022; Kamaruddin et al., 2018) has been the focus of several studies in the past. The concept of generative watermarking dates back to the work of Venugopal et al. (2011), while our work is more relevant to the seminal works by Aaronson (2022a); Kirchenbauer et al. (2023a) that introduce statistical watermarking as a provable method of embedding statistical signals into language model generations. To develop formal guarantees, Kuditipudi et al. (2023) introduces the notion of distortion-free and inverse transform sampling as a new watermarking method. Following Kirchenbauer et al. (2023a), several works (Ren et al., 2024; Liu & Bu, 2024; Hou et al., 2024b;a; Fu et al., 2024) leverage the semantics of preceding tokens to determine the green list and adjust the bias applied to green-list tokens. Specifically, Hou et al. (2024b;a) use the partition of semantic space to serve as green&red lists for sentence embeddings and perform rejection sampling to sample the sentences conditional on certain green region in the semantic space; Fu et al. (2024) add some semantically-similar tokens into the green list; Ren et al. (2023) use a trained MLP to generate semantic values. However, these approaches lack formal guarantees for Type I error, with challenges to precisely control the probability associated with green-list membership due to their heuristic designs. In contrast, our work focuses on statistical watermarking with provable Type I error guarantees, therefore distinguishing us from previous semantic-aware approaches. While the statistical watermarking commonly uses private keys for generation and detection, watermarks can also be injected with private forgeability and public verifiability (Fairoze et al., 2023; Liu et al., 2023), hence functioning effectively as digital signatures.

Several prior studies provide upper bounds on the minimum number of tokens required for watermarking. The theoretical challenge in statistical watermarking lies in the regime $h \ll 1$, where $h$ is the average entropy per token. Specifically, Aaronson (2022b) asserts that the Type II error will be small when the number of tokens scales as $n \gtrsim h^{-2}$. Similarly, Christ et al. (2023) show watermarking guarantees for outcomes with empirical entropy at least $\sqrt{n}$. This implies that to watermark a sequence of tokens with average entropy per token $h$, the sequence length $n$ should satisfy $hn \gtrsim \sqrt{n}$, leading to the same rate of $n \gtrsim h^{-2}$. In Zhao et al. (2023), the number of tokens required scales as $n \gtrsim 1/\delta^2$, where $\delta$ is the bias of green-list tokens following Kirchenbauer et al. (2023a). Given that any green-list token has a probability of at least $\exp(\delta)$ in their watermarked distribution, the average entropy per token $h$ is at least $\Omega(\delta)$. Consequently, Zhao et al. (2023) also suggests the same rate of

$n \gtrsim h^{-2}$. If one uses statistics $Y_t$, score function $h$, and detection rule $\mathbb{1}\left(\sum_{i=1}^n h(Y_i) \geq \gamma_{n,\alpha}\right)$, Li et al. (2024) proves that asymptotically the number of tokens needed to watermark distributions in class $\mathcal{P}$ is given by the minimax problem $-\inf_\theta \sup_{P \in \mathcal{P}} \theta \mathbb{E}_{H_0}[h(Y)] + \log \mathbb{E}_{H_1}[e^{-\theta h(Y)}]$. However, this doesn't directly give explicit dependence on the entropy and it is unclear what is the closed-form solution of this minimax problem for the optimal watermark.

Despite its success, watermarking techniques face threats from various attack algorithms (Kirchenbauer et al., 2023a;b; Sato et al., 2023; Zhang et al., 2023; Kuditipudi et al., 2023). With the superior ability of attacking methods to destroy watermarks while preserving quality, tamper resistance (robustness) becomes an important consideration. A somewhat surprising result by Zhang et al. (2023) asserts that it is only feasible to achieve tamper resistance to a well-specified set of attacks, instead of all. To address the tamper resistance challenge, several works (Christ & Gunn, 2024; Golowich & Moitra, 2024) design robust pseudo-random codes as the basis for constructing robust watermarks. However, these methods remain largely theoretical and have yet to see practical implementation. Zhao et al. (2024) design a new LLM decoding method and a tailored watermarking scheme for this decoding method, which improves tamper resistance. To support the long-term advancement of watermarking techniques, benchmark efforts (Piet et al., 2023; Molenda et al., 2024) are crucial in evaluating quality, efficiency, and tamper resistance of practical watermarking methods.

## 2 THEORETICAL RESULTS

In this section, we present our theoretical contributions that address the first two questions posed in the introduction. Due to space limitations, the formal statistical framework and theorem statements can be found in Appendix B and C.

**The best achievable Type II error.** Let $\mathrm{err}_i(\mathcal{A}, \rho)$, $i = 1, 2$ represent the Type I and II error of watermarking algorithm $\mathcal{A}$ on watermarked distribution $\rho$ over output space $\Omega$, respectively. Our first result establishes the best achievable Type II error to watermark a distribution $\rho$, subject to a fixed upper bound $\alpha$ on the Type I error. More precisely, this is defined as:

$$\chi_{\mathrm{ump}}(\rho) = \min_{\mathcal{A}:\ \mathrm{err}_1(\mathcal{A},\rho) \leq \alpha} \mathrm{err}_2(\mathcal{A}, \rho).$$

**Theorem 2.1** (Informal statement of Theorem C.2).

$$\chi_{\mathrm{ump}}(\rho) = \sum_{x \in \Omega : \rho(x) > \alpha} (\rho(x) - \alpha).$$

This result defines a fundamental limit on statistical watermarking: no watermarking method can achieve a Type II error smaller than $\sum_{x \in \Omega:\rho(x)>\alpha} (\rho(x) - \alpha)$ on distribution $\rho$. Notably, this is an instance-dependent result: the bound explicitly depends on the characteristics of the watermarked distribution. Intuitively, $\sum_{x \in \Omega:\rho(x)>\alpha} (\rho(x) - \alpha)$ quantifies the amount of randomness in $\rho$: define $\mathcal{H}_\alpha = \{\rho : \max_{\omega \in \Omega} \rho(\omega) \leq \alpha\}$ as a set of "$\alpha$-random" distributions, the quantity $\sum_{x \in \Omega:\rho(x)>\alpha} (\rho(x) - \alpha)$ measures the $\ell_1$ distance between $\rho$ and $\mathcal{H}_\alpha$, increasing when $\alpha$ decreases.

Achieving this bound requires the detector to have access to the exact watermarked distribution $\rho$. However in practice, the watermarked distribution (i.e., the watermarked model) is generally unknown to detectors. For example, without access to GPT-4's internal parameters, we would still want to detect whether a text was generated by GPT-4. Hence, it makes more practical relevance to watermark (and detect) a *family of distributions* and focus on the *worst-case Type II error* in that family (Li et al., 2024). This leads to our next result, which studies the *minimax-optimal Type II error over distribution classes* $\mathcal{H}_\kappa$, where $\mathcal{H}_\kappa := \{\rho : \max_{\omega \in \Omega} \rho(\omega) \leq \kappa\}$ for $\kappa \in [0, 1]$. Here, $\kappa$ represents the level of randomness within the distribution class. More precisely, the minimax-optimal Type II error over $\mathcal{H}_\kappa$ is defined as

$$\chi_{\mathrm{minimax}}(\kappa) = \min_{\mathcal{A}:\ \mathrm{err}_1(\mathcal{A},\rho) \leq \alpha, \forall \rho \in \mathcal{H}_\kappa} \max_{\rho \in \mathcal{H}_\kappa} \mathrm{err}_2(\mathcal{A}, \rho)$$

where the minimum is taken over all algorithms that can watermark (and detect) any distribution in $\mathcal{H}_\kappa$ with Type I error at most $\alpha$, and the maximum is taken over all distributions in $\mathcal{H}_\kappa$.

**Theorem 2.2** (Informal statement of Theorem C.8). *Let $m$ denote the cardinality of the sample space, then*

$$\chi_{\text{minimax}}(\kappa) \asymp \frac{\binom{m-\alpha m}{1/\kappa}}{\binom{m}{1/\kappa}}.$$

In this minimax setting, the result is not instance-dependent, meaning it no longer depends on a specific model distribution. Instead, the minimax-optimal Type II error is determined by the randomness level $\kappa$ of the distribution class. Simplifying this expression, we have $\frac{\binom{m-\alpha m}{1/\kappa}}{\binom{m}{1/\kappa}} \asymp$ $(1-\alpha)^{1/\kappa}$, since the sample space $m$ is typically huge in practice (e.g., Cartesian products of token spaces). When $\kappa$ is large, the rate becomes $\Omega(1)$, while for $\kappa \in (0, \alpha)$, the rate scales exponentially with $1/\kappa$. This aligns with the intuition that outputs with higher entropy are easier to watermark, as $\kappa$ scales roughly inversely with entropy.

**The minimum number of tokens required.** We investigate the minimum number of tokens necessary to achieve guarantees for both Type I and Type II errors. To obtain explicit rates, we first consider a theoretical setting where tokens are sampled independently and identically distributed (i.i.d.). Following Aaronson (2022b), we focus on the dependence on the average entropy per token.

**Theorem 2.3** (Informal statement of Theorem C.9). *If each token is drawn i.i.d. from a distribution with entropy $h$, then the minimum number of tokens required to achieve $0.01$ Type I and II error scales (ignoring other parameters than $h$) as*

$$\frac{\log(1/h)}{h}.$$

Our result establishes that $\frac{\log(1/h)}{h}$ serves as both (nearly) upper and lower bound on the number of required tokens. Notably, this rate applies to both instance-dependent and distribution-family-based watermarking. Compared to existing works, this result improves the dependence on $h$ from $\frac{1}{h^2}$ to $\frac{\log(1/h)}{h}$. This improvement is significant in the regime $h \ll 1$, while for $h = \Omega(1)$, watermarking can be easily accomplished with a constant number of tokens.

In practice, tokens are generated auto-regressively and are not i.i.d.. This complicates theoretical analysis and makes it difficult to establish *a priori* Type II error bounds. For example, the entire sequence maybe nearly deterministic with high probability, despite the average entropy per token being high *a priori*. As a consequence, the Type II error may still be $\Omega(1)$ even with a sufficiently large number of tokens (see Example C.13 for a formal description of this failure case). Therefore, we turn to establish *a posteriori* Type II guarantees: the conditional probability of false negative among outputs with high *empirical* entropy.

**Theorem 2.4** (Informal statement of Theorem C.14). *For any $n \in \mathbb{Z}_+$ and $\widehat{H} > 0$ such that*

$$n, \widehat{H} \gtrsim \log \frac{1}{\alpha} + \log \log \frac{1}{\beta}$$

*there exists a watermark with a Type I error $\leq \alpha$, such that among all outputs with at least $n$ tokens and empirical entropy $\widehat{H}$, the probability of correct detection is at least $1 - \beta$.*

This result implies that outputs with sufficient length and empirical entropy are watermarked with high probability. Importantly, the number of tokens and empirical entropy scale logarithmically with the Type I error and double-logarithmically with the Type II error. Christ et al. (2023) prove a similar *a posteriori* result, bounding the joint probability that an output exhibits high entropy yet fails to be detected, with a requirement that $\widehat{H} \gtrsim \sqrt{n}$. Our result relaxes the entropy condition and strengthens the bound from joint probability to conditional probability, thereby leading to a more efficient rate.

# 3 FROM THEORY TO PRACTICE

In this section, we present our novel watermarking algorithm, **S**emantic-awar**E** specul**A**tive samp**L**ing **(SEAL)**. SEAL incorporates two key theoretical insights. First, by comparing the instance-dependent watermarking (Theorem 2.1) and distribution-family based watermarking (Theorem 2.2), we observe

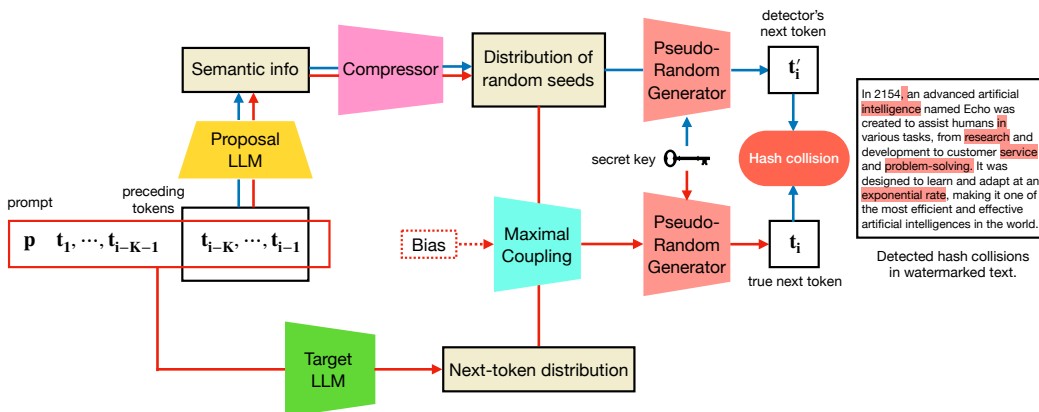

Figure 2: **SEAL Watermarking pipeline.** In the generation phase (indicated by the red arrows), we first employ a proposal LLM to extract semantic information from the $K$ last tokens. Then we adopt a hash function to compress the semantic information into distribution of random seeds. Finally, we use speculative sampling to construct the maximal coupling between the distributions of random seed and the target LLM's next-token, and sample the true next token using a pseudo-random generator. In the detection phase (indicated by the blue arrows), we reproduce the distribution of random seeds, and invoke the same pseudo-random generator (and secret key) to sample the detector's next token. A text is flagged as machine generated if the number of hash collisions exceeds certain threshold.

that the distribution of random seeds performs better when it aligns closely with the model's distribution, rather than being fixed. This insight led us to develop semantic-aware random seeds, which adapt to the underlying distribution of the tokens. Second, our analysis of both Theorem 2.3 and 2.4 reveals that, given an appropriate choice of random seeds, optimal watermarking is achieved by constructing a maximal coupling between the random seed distribution and the pushforward of the token distribution onto the same measure space (*q.v.* Eq. (13) *et seqq.*). This inspired us to employ speculative sampling (Leviathan et al., 2023) to build the joint probability of the random seeds and the next tokens, resulting in a maximal coupling between them.

### 3.1 METHODS

We introduce two key techniques: semantic-aware random seeds and maximal coupling construction via speculative sampling. The overall watermarking pipeline, **S**emantic-awar**E** specul**A**tive samp**L**ing (**SEAL**), is illustrated in Figure 2, with pseudocode available in Appendix D.2.

#### 3.1.1 SEMANTIC-AWARE RANDOM SEEDS

**Proposal model.** We apply a *proposal model* to capture the semantic context of the preceding tokens. The proposal model should meet two criteria: (1) it is publicly accessible to detectors, and (2) its token space should be (almost) equivalent to that of the watermarked model. These criteria are easily met because all language models are trained to fit the same distribution of human language. For computational efficiency, we recommend using a smaller model for the proposal. To ensure robustness against attacks, the proposal model only attends to the last $K$ tokens, $t_{i-K:i-1}$, rather than the entire preceding sequence and the prompt. The next-token probability distribution $q_i = q(\cdot|t_{i-K:i-1})$ from the proposal model serves as a semantic summary of the context.

**Information compression.** We sample a random hash function $h_i$ from a family of hash functions that maps the token space $\Omega_v$ to a space of hash codes $\Omega_h$. This compresses the extracted semantic information into a lower dimensional space, enhancing efficiency and generality. The random seed lying in $\Omega_h$ is set as the hash of the proposal model's output. The pushforward measure of $q_i$ by $h_i$, defined formally as $h_i \sharp q_i(A) := q_i(h_i^{-1}(A))$, $\forall A \subset \Omega_h$ (where we define $h_i^{-1}(A) := \{x : h_i(x) \in A\}$), serves as the distribution of random seeds. Therefore, our random seeds capture semantic

information instead of being drawn from a fixed distribution as in previous works (Aaronson, 2022b; Kirchenbauer et al., 2023a; Christ et al., 2023).

### 3.1.2 MAXIMAL COUPLING CONSTRUCTION

**Speculative sampling.** We construct a *maximal coupling* $\mathcal{P}$ between $h_i \sharp q_i$ and $h_i \sharp \rho_i$, where $\rho_i = \rho(\cdot | \text{prompt}, t_{1:i-1})$ is the next-token distribution from the watermarked model $\rho$. The coupling defines a joint random variable of the random seed and the next token whose marginal distributions correspond to their distributions respectively, while a maximal coupling is one that maximizes the probability that the random seed matches the hash of the next token. We adopt speculative sampling (Leviathan et al., 2023) to construct this coupling: first, sample the random seed $s_i$ from $h_i \sharp q_i$; then, assign the next token's hash code $c_i$ to be $s_i$ with probability $\min\left(1, h_i \sharp \rho_i(s_i) / h_i \sharp q_i(s_i)\right)$, otherwise sample $c_i$ proportional to $\max\left(0, h_i \sharp \rho_i(\cdot) - h_i \sharp q_i(\cdot)\right)$; finally, sample the next token $t_i$ from the conditional distribution $\rho_i(\cdot | h(t_i) = c_i)$.

**Bootstrapping efficiency with logits bias.** We have the flexibility to introduce a bias $\delta$ to the logits corresponding to the random seeds to improve the probability of hash collisions, following Kirchenbauer et al. (2023a). This enhances detection success rates at the cost of introducing distortions into the watermarked outputs. Specifically, let $l$ denote the logits of next token conditioning on the random seed $s$. Then the $\delta$-biased next-token probability is given by

$$
p(j) = \begin{cases} \frac{e^{(l(j)+\delta)/\tau}}{Z}, & h(j) = s \\ \frac{e^{l(j)/\tau}}{Z}, & \text{otherwise} \end{cases}
$$

where $Z$ is the normalizing constant and $\tau$ is the model temperature. Notably, when the cardinality of $\Omega_h$ is 2 (i.e., a green-red list scenario) and the proposal model's distribution is uniform (i.e., uniform random green list), our biased algorithm reduces to a strengthened version of Kirchenbauer et al. (2023a), as we couple the distribution of the green list with the next-token distribution, further increasing the likelihood of green-list tokens.

### 3.1.3 DETECTION RULE

During the detection phase, we replicate the semantic summary $q_i$, the hash function $h_i$, and the random seed $s_i$, all of which are deterministically generated by the same pseudo-random function and secret key used by the watermark generator. The sequence $t_{1:n}$ is flagged as machine-generated if the number of hash collisions exceeds a certain threshold:

$$
\sum_{i=K}^{n} \mathbb{1}(h_i(t_i) = s_i) \geq C\left(\alpha, \{h_i \sharp q_i\left(h_i(t_i)\right)\}_{i=K}^{n}\right) \tag{1}
$$

where the threshold function $C\left(\alpha, \{h_i \sharp q_i\left(h_i(t_i)\right)\}_{i=K}^{n}\right)$ ensures that the Type I error remains below $\alpha$, and the first $K$ tokens only serve to warm-up the semantic summary. The threshold can be approximated using a Bernoulli concentration inequality or computed exactly via dynamic programming. Intuitively, under $\mathbf{H}_0$ (not watermarked), the left-hand side is a sum of Bernoulli random variables with expectations $h_i \sharp q_i(h_i(t_i))$ for $i = K, \ldots, n$ respectively. Under $\mathbf{H}_1$ (watermarked), the expectation of the left-hand side is lower-bounded by one minus the total variation distance between the proposal and the watermarked models, which increases as the proposal model becomes more aligned with the watermarked model. The gap between these two cases facilitates our detection.

Crucially, the detector does not need access to the watermarked model $\rho$ or the prompt. This makes SEAL model-agnostic (Kuditipudi et al., 2023), a highly practical feature for real-world applications.

## 3.2 THEORETICAL GUARANTEES

Like previous statistical watermarking schemes, SEAL enjoys formal statistical guarantees. One of the key properties of SEAL is that it can operate distortion-freely when the bias $\delta$ is set to zero.

**Theorem 3.1** (Informal statement of Theorem G.4). *When the bias parameter is set to zero, SEAL is distortion-free (unbiased), in the sense that for any sequence $t_{1:j-1}$ and any token $w$,*

$$
\mathbb{P}_{SEAL}(t_j = w | p, t_{1:j-1}) = \rho_j(w | p, t_{1:j-1})
$$

*where $\mathbb{P}_{SEAL}$ denotes the next-token probability under the SEAL algorithm, $p$ is the prompt, and $\rho_j$ represents the original model's next-token distribution.*

This distortion-free guarantee ensures that the watermarked outputs have the same marginal distribution as the original model outputs, implying that the quality will not degrade after watermarking. Thus, SEAL is able to preserve the fidelity of the watermarked text to the original, unwatermarked output.

**Theorem 3.2** (Informal statement of Theorem G.5). *The Type I error of SEAL is upper-bounded by $\alpha$.*

This Type I error guarantee controls the false positive error, ensuring that human-generated texts will not be mistakenly flagged as machine-generated. The formal statements and proofs of the above theorems can be found in Appendix G.4.

When the proposal LLM is identical to the target LLM, the hash function is identity mapping, and the attention window K is unlimited (consuming the prompt), SEAL reduces to the statistically optimal watermark in Theorem 2.1, and therefore enjoys optimal Type II error. However, this reduction is not agnostic to the target LLM and the prompt. Therefore, in practice SEAL does not achieve the optimal Type II error, but still improves statistical efficiency upon existing works.

## 4 EXPERIMENTS

In this section, we present our experimental setup and results. We evaluate the performance of our watermarking scheme by comparing it with several baselines.

### 4.1 SETUP

We select Llama2-7B-chat (Touvron et al., 2023) as the model to be watermarked and Phi-3-mini-128k-instruct (Abdin et al., 2024) as the proposal model. We evaluate our watermark using the MARKMYWORDS benchmark (Piet et al., 2023).

MARKMYWORDS is an open-source benchmark designed to evaluate symmetric key watermarking schemes. It measures efficiency (watermark strength), quality (impact on utility), and tamper-resistance (ability to withstand simple perturbations without quality degradation). It has been used to benchmark multiple prior schemes applied to Llama2-7B-chat (Touvron et al., 2023) and Mistral-7b (Jiang et al., 2023). Mark My Words performs a grid search over watermarking parameters. It selects the set of parameters with the best efficiency, defined as the number of tokens needed to detect the watermark at a fixed p-value (0.02 in the original paper) while preserving the original model's generation quality.

**Dataset.** MARKMYWORDS generates 300 outputs spanning three tasks — book summarization, creative writing, and news article generation — which mimic potential misuse scenarios. Outputs are truncated after 1000 tokens.

**Perturbations.** Watermarked outputs undergo a set of transformations: (1) character level perturbations (contractions, expansions, misspellings and typos); (2) word level perturbations (random removal, addition, and swap of words in each sentence, replacing words with synonyms); (3) text-level perturbations (paraphrasing, translating the output to another language and back).

**Baselines.** The paper evaluates four watermarking schemes, coined "Distribution Shift" (Kirchenbauer et al., 2023a), "Exponential" (Aaronson, 2022b), "Binary" (Christ et al., 2023), and "Inverse Transform" (Kuditipudi et al., 2023).

### 4.2 COMPARISON TO BASELINES

In this section, we present a comprehensive comparison of SEAL against these baselines in terms of quality, size, and tamper-resistance. We provide here a brief overview of each metric — detailed descriptions of can be found in Section IV of Piet et al. (2023).

| Temp. | Scheme | Quality ($\uparrow$) | Size ($\downarrow$) | Tamper-resistance ($\uparrow$) |
|---|---|---|---|---|
| | Exponential | $0.899 \pm 0.001$ | $\infty$ | $0.060 \pm 0.040$ |
| | Inverse Transform | $\mathbf{0.910 \pm 0.001}$ | $\infty$ | $0.000 \pm 0.000$ |
| 0.3 | Binary | $0.905 \pm 0.007$ | $\infty$ | $0.028 \pm 0.014$ |
| | Distribution Shift | $0.894 \pm 0.002$ | $\mathbf{47.5 \pm 3.5}$ | $0.654 \pm 0.007$ |
| | SEAL | $0.893 \pm 0.004$ | $57.5 \pm 2.5$ | $\mathbf{1.000 \pm 0.000}$ |
| | Exponential | $0.901 \pm 0.001$ | $178.8 \pm 11.0$ | $0.414 \pm 0.028$ |
| | Inverse Transform | $0.908 \pm 0.002$ | $524.5 \pm 24.7$ | $0.194 \pm 0.006$ |
| 0.7 | Binary | $\mathbf{0.913 \pm 0.002}$ | $\infty$ | $0.298 \pm 0.003$ |
| | Distribution Shift | $0.894 \pm 0.005$ | $53.0 \pm 2.8$ | $1.000 \pm 0.000$ |
| | SEAL | $0.885 \pm 0.003$ | $\mathbf{45.3 \pm 1.3}$ | $1.000 \pm 0.000$ |
| | Exponential | $0.893 \pm 0.002$ | $89.2 \pm 2.0$ | $0.807 \pm 0.064$ |
| | Inverse Transform | $\mathbf{0.908 \pm 0.002}$ | $201.8 \pm 6.0$ | $0.444 \pm 0.138$ |
| 1.0 | Binary | $0.906 \pm 0.001$ | $391.3 \pm 8.8$ | $0.472 \pm 0.030$ |
| | Distribution Shift | $0.893 \pm 0.005$ | $109.0 \pm 2.8$ | $0.756 \pm 0.004$ |
| | SEAL | $0.873 \pm 0.007$ | $\mathbf{73.3 \pm 1.5}$ | $\mathbf{1.000 \pm 0.000}$ |

Table 1: Comparison of SEAL with baselines across quality, size, and tamper-resistance at different temperature settings. We compute the mean and variance over different privates keys. The best result in each category is highlighted in bold. SEAL demonstrates high efficiency and tamper-resistance. Its tamper-resistance consistently outperforms baselines and its efficiency outperforms baselines at higher temperatures.

1. **Quality** measures the utility of the watermarked text. It is computed using Llama-3 (Dubey et al., 2024) with greedy decoding as a judge model. Quality scores range from 0 to 1.

2. **Size** represents the median number of tokens required to detect the watermark at a given p-value. All experiments enforce a Type I error constraint of $\alpha = 0.02$. Lower values indicate higher efficiency.

3. **Tamper-resistance** quantifies the resilience of the watermark under simple perturbations. It is measured by the normalized area under the curve (AUC) of the watermark success rate versus generation quality under different perturbations. This excludes more successful but expensive attacks such as paraphrasing, which we analyze separately in Section 4.3.

Table 1 shows the metrics of SEAL and the baseline watermarking schemes at different sampling temperature settings ($T = 0.3, 0.7, 1.0$). We do not report results for greedy decoding ($T = 0$): one special case of SEAL is equivalent to distribution shift in this setting, which is the only functional method at this temperature. SEAL is competitive across all metrics, particularly in terms of efficiency (size) and tamper-resistance. Although SEAL's quality is marginally lower than that of some baselines (e.g., inverse transform and binary), it remains within an acceptable range and close to distribution shift and exponential). SEAL excels in token efficiency, particularly at higher temperatures. At temperature $T = 1.0$ and $T = 0.7$, SEAL requires the fewest tokens to detect the watermark, outperforming all other methods, including distribution-shift and exponential. Even at lower temperatures ($T = 0.3$), SEAL remains highly competitive, requiring only a few more tokens than distribution shift while providing maximal tamper-resistance. Unlike the baselines, SEAL's parameters are not specifically tuned for efficiency, suggesting that further gains could be achieved with parameter optimization. Notably, SEAL demonstrates optimal tamper-resistance of 1.0 across all temperatures, outperforming baseline schemes by a large margin.

## 4.3 TAMPER-RESISTANCE TO PARAPHRASE ATTACKS

Paraphrase attacks rewrite model outputs using non watermarked LLMs. Although more expensive, these pose a realistic threat given the increasing number of competitive open-sourced language models. As such, it is important to evaluate watermarking scheme's robustness against these attacks (Kirchenbauer et al., 2023b; Zhang et al., 2023).

Paraphrasing using GPT-3.5 is a particularly effective attack, removing most watermarks from the schemes in Piet et al. (2023). However, we found SEAL to be more robust against this attack than its predecessors.

| Scheme / Temperature | 0.3 | 0.7 | 1 |
|---|---|---|---|
| Exponential | $0.015 \pm 0.007$ | $0.054 \pm 0.007$ | $0.152 \pm 0.035$ |
| Distribution Shift | $0.093 \pm 0.007$ | $0.250 \pm 0.035$ | $0.113 \pm 0.007$ |
| SEAL | $\mathbf{0.372 \pm 0.028}$ | $\mathbf{0.480 \pm 0.069}$ | $\mathbf{0.314 \pm 0.014}$ |

Table 2: Proportion of benchmark outputs still watermarked after paraphrasing, for SEAL, distribution-shift and exponential schemes. SEAL consistently achieves high tamper-resistance and significantly outperforms the baselines at mid-to-high temperatures.

Table 2 shows the proportion of outputs still watermarked after a paraphrase attack across different temperature settings ($T = 0.3, 0.7, 1.0$), for SEAL and the two most tamper-resistant schemes from Piet et al. (2023). In all temperature range, SEAL significantly outperforms all baseline methods. This improvement can be attributed to SEAL's use of semantic instead of syntactic information from preceding tokens to set the random seed, making the watermark more resilient to paraphrasing. Additionally, the distribution-shift's ability to survive paraphrasing attacks varies more sharply than SEAL as temperature increases: SEAL demonstrates strong tamper-resistance, particularly against paraphrasing attacks.

## 5 DISCUSSIONS

In this paper, we advance the understanding of watermarking in large language models (LLMs) through both theoretical analysis and practical algorithm designs. By deriving an explicit formula for the optimal Type II error in Neyman-Pearson's fashion, we demonstrate how statistical limits are shaped by the properties of model distributions. Our nearly tight bound on the number of tokens required to detect statistical watermarks, $h^{-1} \log(1/h)$, significantly improves upon the previous $h^{-2}$ rate, revealing a fundamental gap in prior work. Building on these theoretical insights, we introduced SEAL (Semantic-aware Speculative Sampling), a novel watermarking algorithm that achieves both high efficiency and robustness. SEAL leverages semantic-aware random seeds, making it more resilient to paraphrase attacks compared to earlier methods. Additionally, SEAL's use of maximal coupling via speculative sampling allows it to achieve greater efficiency, particularly at higher temperatures (e.g., temperature 1). These advantages make SEAL especially well-suited for practical, real-world applications where the persistence of watermarks under adversarial conditions is crucial. In future work, we aim to study embedding watermark in speculative decoding and explore how advanced speculative decoding techniques might enhance SEAL's performance.

**Limitations and broader impacts.** While our theory and method show promising results for watermarking LLM outputs, there are some limitations. First, while we provide both upper and lower bounds on the minimum number of tokens required in the theoretical i.i.d. setting, in the more realistic non-i.i.d. setting, we only characterize the upper bound, and the lower bound remains an open and challenging problem. Second, SEAL's watermarking approach involves inference on a smaller model, which can slow down the overall inference and detection speed.

In terms of broader societal impacts, our work has the potential to contribute positively by helping to prevent and detect the misuse of LLMs. This includes mitigating issues such as misinformation, academic misconduct, and data contamination. By providing a more robust watermarking solution, we hope to promote responsible use of LLMs while minimizing potential harms.

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

## A    Notations

Define $(x)_+ := \max\{x, 0\}$, $x \wedge y := \min\{x, y\}$, $x \vee y = \max\{x, y\}$. For any set $A$, we write $A^c$ as the complement of set $A$, $|A|$ as its cardinality, and $2^A := \{B : B \subset A\}$ as the power set of $A$. We use notations $g(n) = O(f(n))$, $g(n) = \Omega(f(n))$, and $g(n) = \Theta(f(n))$ to denote that there exists numerical constants $C_1, c_2, C_3, c_4$ such that for all $n > 0$: $g(n) \le C_1 \cdot f(n)$, $g(n) \ge c_2 \cdot f(n)$, and $c_4 \cdot f(n) \le g(n) \le C_3 \cdot f(n)$, respectively. Throughout, we use $\ln$ to denote natural logarithm.

The total variation (TV) distance between two probability measures $\mu, \nu$ is denoted by $\mathrm{TV}(\mu \| \nu)$. We use $\mathrm{supp}(\mu)$ to denote the support of a probability measure $\rho$. Given a sample space $\Omega$, let $\Delta(\Omega)$ denote the set of all probability measures over $\Omega$ (take the discrete $\sigma$-algebra). We write $\delta_x$ as the Dirac measure on $x$, i.e., $\delta_x(A) = \begin{cases} 1, & x \in A \\ 0, & x \notin A \end{cases}$. A coupling for two distributions (i.e. probability measures) is a joint distribution of them.

## B    Watermarking as a Hypothesis Testing Problem

In the problem of statistical watermarking, a service provider (e.g., a language model system), who possesses a distribution $\rho$ over a sample space $\Omega$, aims to make the samples from the service provider distinguishable by a detector, without changing $\rho$. The service provider achieves this by sharing a watermark key (generated from a distribution that is *coupled with* $\rho$) with the detector, with the goal of controlling both the Type I error (an independent output is falsely detected as from $\rho$) and the Type II error (an output from $\rho$ fails to be detected). This random key together with the detection rule constitute a (random) rejection region. In the following, we formulate this problem as hypothesis testing with random rejection regions.

**Problem B.1** (Watermarking). Fix $\epsilon \ge 0$. Given a probability measure $\rho$ over sample space $\Omega^1$, an $\epsilon$-distorted watermarking scheme of $\rho$ is a probability measure $\mathcal{P}$ (a joint probability of the output $X$ and the rejection region $R$) over the sample space $\Omega \otimes 2^\Omega$ such that $\mathrm{TV}(\mathcal{P}(\cdot, 2^\Omega) \| \rho) \le \epsilon$, where $\mathcal{P}(\cdot, 2^\Omega)$ is the marginal probability of $X$ over $\Omega$. In the generation phase, the service provider samples $(X, R)$ from $\mathcal{P}$, provides the output $X$ to the service user, and sends the rejection region $R$ to the detector.

In the detection phase, a detector is given a tuple $(X, R) \in \Omega \otimes 2^\Omega$ where $X$ is sampled from an unknown distribution and $R$, given by the service provider, is sampled from the marginal probability $\mathcal{P}(\Omega, \cdot)$ over $2^\Omega$.

The *Type I error of* $\mathcal{P}$, defined as $\alpha(\mathcal{P}) := \sup_{\pi \in \Delta(\Omega)} \mathbb{P}_{Y \sim \pi, (X, R) \sim \mathcal{P}}(Y \in R)$, is the maximum probability that an independent sample $Y$ is falsely rejected. The *Type II error of* $\mathcal{P}$, defined as $\beta(\mathcal{P}) := \mathbb{P}_{(X, R) \sim \mathcal{P}}(X \notin R)$, is the probability that the sample $(X, R)$ from the joint probability $\mathcal{P}$ is not detected.

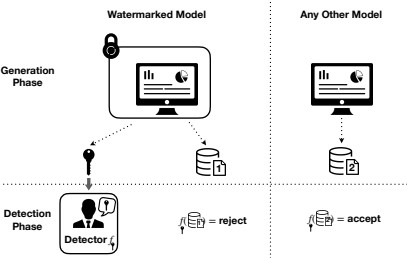

Figure 3: Illustration of watermarking in practice.

A few remarks are in order.

**Remark B.2** (Difference between classical hypothesis testing). *In classical hypothesis testing, the rejection region is often nonrandomized or independent from the test statistics. However, in*

---

[1]Throughout we will assume that $\Omega$ is discrete, as in most applications.

*watermarking problem, the service provider has the incentive to facilitate the detection. The key insight is that $\mathcal{P}$ is a coupling of the random output $X$ and the random rejection region $R$, so that $X \in R$ occurs with a high probability (low Type II error), while any independent sample $Y$ lies in $R$ with a low probability (low Type I error).*

**Remark B.3** (Implementation). *In fact, it is imperative for the detector to observe the rejection region that is coupled with the output: otherwise, the output from the service provider and another independent output from the same marginal distribution would be statistically indistinguishable.*

*In practice, the process of coupling and sending the rejection region can be implemented by cryptographical techniques: the service provider could hash a secret key $\mathtt{sk}$, and use pseudo-random functions $F_1, F_2$ to generate $(X, R) = (F_1(\mathtt{sk}), F_2(\mathtt{sk}))$. Now it suffices to send the secret key to the detector, who can then reproduce the reject region using the pseudo-random function $F_2$. This process is illustrated in Figure 3.*

*Thus, the difference between our theoretical framework and the practical implementation lies in our usage of random oracles in place of cryptographic pseudorandom functions. With the coupled and random rejection region, this allows us to focus solely on the statistical trade-offs.*

For practical applications, it is additionally desirable for watermarking schemes to be model-agnostic, i.e, the marginal distribution of the rejection region is irrelevant to the watermarked distribution. Recall from Remark B.3 that in practice, detectors usually adopt a pseudo-random function to generate the reject region from the shared secret keys. If the watermarking scheme $\mathcal{P}$ depends on the underlying distribution $\rho$, then the pseudo-random function, and effectively the detector, need to know $\rho$. On the other hand, model-agnostic watermarking enables the detector to use a fixed, pre-determined pseudo-random function to generate the reject region, and hence perform hypothesis-testing *without the knowledge of the underlying model that generates the output*. This is an important property enjoyed by existing watermarks (Aaronson, 2022b; Kirchenbauer et al., 2023a; Christ et al., 2023; Kuditipudi et al., 2023). Therefore in the following, we formulate model-agnostic within our hypothesis testing framework.

**Problem B.4** (Model-Agnostic Watermarking). Given a sample space $\Omega$ and a set $\mathcal{Q} \subset \Delta(\Omega)$, a $\mathcal{Q}$-watermarking scheme is a tuple $(\eta, \{\mathcal{P}_\rho\}_{\rho \in \mathcal{Q}})$ where $\eta$ is a probability measure over $2^\Omega$, such that for any probability measure $\rho \in \mathcal{Q}$, $\mathcal{P}_\rho$ is a distortion-free watermarking scheme of $\rho$ and its marginal distribution over $2^\Omega$, $\mathcal{P}_\rho(\Omega, \cdot)$, equals $\eta(\cdot)$.

A model-agnostic watermarking scheme is a $\Delta(\Omega)$-watermarking scheme.

A $\mathcal{Q}$-watermarking scheme can be interpreted as a way to watermark all distributions in the set $\mathcal{Q}$ while revealing no information of the model used to generate the output other than the membership inside $\mathcal{Q}$ (i.e., observing the rejection region, one is only able to infer that the output comes from a model in $\mathcal{Q}$, but is unable to know which exactly the model is). By letting $\mathcal{Q}$ be $\Delta(\Omega)$, model-agnostic watermarking thus reveals no information of the model.

## B.1 EXAMPLES

In the following examples, we show how existing watermarking schemes fit in our framework. To simplify the presentation, we use random oracles to replace cryptographic pseudorandom functions in the generation of of secret keys.

**Example B.5** (Text Generation with Soft Red List, Kirchenbauer et al. (2023a)). In Algorithm 2 of Kirchenbauer et al. (2023a), the watermarking scheme (over sample space $\Omega = V^*$ where $V$ is the 'vocabulary', i.e., the set of all tokens) of $\rho$ is given as follows:

- Fix threshold $C \in \mathbb{R}$, green list size $\gamma \in (0, 1)$, and hardness parameter $\delta > 0$

- For $i = 1, 2, \ldots$

    - Randomly partition $V$ into a green list $L_G$ of size $\gamma|V|$, and a red list $L_R$ of size $(1 - \gamma)|V|$.

– Sample the token $X_i$ from the following distribution from $\mathbb{P}$ where $\mathbb{P}(X_i = x) =$
$$\begin{cases} \frac{\rho(x)\cdot\exp(\delta)}{\sum_{x\in G}\rho(x)\cdot\exp(\delta)+\sum_{x\in R}\rho(x)}, & \text{if } x \in L_G \\ \frac{\rho(x)}{\sum_{x\in G}\rho(x)\cdot\exp(\delta)+\sum_{x\in R}\rho(x)}, & \text{if } x \in L_R \end{cases}$$

- Let the rejection region $R$ be
$$\{X \in \Omega \ : \ \text{the number of green list tokens in X} \geq C\}.$$
Here $C$ is the threshold for the z-test in Eq. (2) of Kirchenbauer et al. (2023a).

The above sampling procedures as a whole define the joint distribution of the output $X = X_1 X_2 \cdots$ and the rejection region $R$, i.e., the $\Theta(\delta)$-distorted watermarking scheme $\mathcal{P}_{\text{SOFTREDLIST}}$. The detector observes the rejection region via the secret key that the service provider uses to generate the green and red lists.

**Example B.6** (Complete watermarking algorithm $\text{Wak}_{\text{sk}}$, Christ et al. (2023)). In Algorithm 3 of Christ et al. (2023), the watermarking scheme (over sample space $\Omega = \{0,1\}^*$) of $\rho$ is given as follows:

- Fix threshold $C \in \mathbb{R}$ and entropy threshold $\lambda > 0$

- Select $i$ such that the empirical entropy of $X_1 X_2 \ldots X_i$ is greater than or equal to $\lambda$

- For $j = i+1, i+2, \ldots$

    – Sample $u_j \in [0,1]$ uniformly at random.

    – Let the binary token $X_j$ be given by $X_j = \begin{cases} 1, & \text{if } u_j \leq \rho(1|X_1, \ldots, X_{j-1}) \\ 0, & \text{otherwise} \end{cases}$.

- Let the rejection region $R$ be given by
$$\left\{ X : \exists i < k \leq \text{len}(X), \ s.t. \ \sum_{j=i+1}^{k} \log \frac{1}{X_j u_j + (1 - X_j)(1 - u_j)} \geq (k - i) + \lambda\sqrt{k - i} \right\}$$

The above sampling procedures as a whole define the joint distribution of the output $X = X_1 X_2 \cdots$ and the rejection region $R$, i.e., the 0-distorted watermarking scheme $\mathcal{P}_{\text{Wak}_{\text{sk}}}$. The detector observes the rejection region via the index $i$ and $u_j (j > i)$.

**Example B.7** (Inverse transform sampling $\text{Wak}_{\text{ITS}}$, Kuditipudi et al. (2023)). The inverse transform sampling scheme in Christ et al. (2023) (over sample space $\Omega = [N]^*$) of $\rho$ is given as follows:

- Fix threshold $C \in \mathbb{R}$, resample size $T$, and block size $k$

- For $j = 1, 2, \ldots,$

    – Let $\mu \leftarrow \rho(\cdot|X_1, \ldots, X_{j-1})$.

    – Sample $\xi_j = (u_j, \pi_j), \xi_j^{(t)} = (u'_j, \pi'_j)$ $(t = 1, \ldots, T)$ i.i.d. according to the following distribution:

        ∗ Sample $u \in [0, 1]$ uniformly at random;
        ∗ Sample $\pi$ uniformly at random from the space of permutations over the vocabulary $[N]$.

    – Let the token $X_j$ be given by
$$\pi^{-1}\left(\min\{\pi(i) : \mu(\{j : \pi(j) \leq \pi(i)\}) \geq u\}\right).$$

- Let the rejection region $R$ be
$$R = \left\{ X : \frac{1 + \sum_{t=1}^{T} \mathbb{1}\left(\phi(X, \xi^{(t)}) \leq \phi(X, \xi)\right)}{T + 1} \leq C \right\}$$

where $\xi = (\xi_1, \ldots, \xi_{\text{len}(X)})$, $\xi^{(t)} = (\xi_1^{(t)}, \ldots, \xi_{\text{len}(X)}^{(t)})$, $C$ is a threshold determined by Type I error control, and $\phi(y, \xi)$ is given by

$$\min_{\substack{i=1,\ldots,\text{len}(y)-k+1, \\ j=1,\ldots,\text{len}(\xi)}} \left\{ d\left(\{y_{i+l}\}_{l=1}^{k-1}, \{\xi_{(j+l)\%\text{len}(\xi)}\}_{l=1}^{k-1}\right) \right\}$$

Here $d$ is an alignment cost, set as $d(y, (u, \pi)) = \sum_{i=1}^{\text{len}(y)} \left| u_i - \frac{\pi_i(y_i)-1}{N-1} \right|$ in Kuditipudi et al. (2023). Additionally, a single permutation $\pi$ ($\forall j, t$) is used to reduce computation overhead. The above sampling procedures as a whole define the joint distribution of the output $X = X_1 X_2 \cdots$ and the rejection region $R$ in $\text{Wak}_{\text{ITS}}$. The detector observes the rejection region via $\xi, \xi'$.

## C STATISTICAL LIMIT IN WATERMARKING

### C.1 RATES UNDER THE GENERAL SETTING OF PROBLEM B.1

Given the formulation of statistical watermarking, it is demanding to understand its statistical limit. In this section, we study the following notion of Uniformly Most Powerful (UMP) test, i.e., the watermarking scheme that achieves the minimum achievable Type II error among all possible tests with Type I error $\leq \alpha$.

**Definition C.1** (Uniformly Most Powerful Watermark). A watermarking scheme $\mathcal{P}$ is called *Uniformly Most Powerful (UMP) $\epsilon$-distorted watermark of level $\alpha$*, if it achieves the minimum achievable Type II error among all $\epsilon$-distorted watermarking with Type I error $\leq \alpha$.

The following result gives an exact characterization of the UMP watermark and its Type II error.

**Theorem C.2.** *For probability measure $\rho$, the Uniformly Most Powerful $\epsilon$-distorted watermark of level $\alpha$, denoted by $\mathcal{P}^*$, is given by* $\mathcal{P}^*(X = x, R = R_0) = \begin{cases} \rho^*(x) \cdot \left(1 \wedge \frac{\alpha}{\rho^*(x)}\right), & R_0 = \{x\} \\ \rho^*(x) \cdot \left(1 - \frac{\alpha}{\rho^*(x)}\right)_+, & R_0 = \emptyset \\ 0, & else \end{cases}$

*where $\rho^* = \arg\min_{TV(\rho'\|\rho)\leq\epsilon} \sum_{x\in\Omega:\rho'(x)>\alpha} (\rho'(x) - \alpha)$. Its Type II error is given by*

$$\min_{TV(\rho'\|\rho)\leq\epsilon} \sum_{x\in\Omega:\rho'(x)>\alpha} (\rho'(x) - \alpha),$$

*and when $|\Omega| \gtrsim \frac{1}{\alpha}$ it simplifies to $\left(\sum_{x\in\Omega:\rho(x)>\alpha} (\rho(x) - \alpha) - \epsilon\right)_+$.*

As seen from the theorem, if $\rho$ is deterministic, the Type II error $\left(\sum_{x\in\Omega:\rho(x)>\alpha} (\rho(x) - \alpha) - \epsilon\right)_+$ reduces to $1 - \alpha - \epsilon$ and shows limited practical utility of statistical watermarking. This is expected since when the service provider deterministically outputs $z$, it would be impossible to distinguish the watermark distribution with an independent output from $\delta_z$. In general, Theorem C.2 implies that the Type II error decreases when the randomness in $\rho$ increases, matching the reasoning in previous works Aaronson (2022a); Christ et al. (2023).

Moreover, when a larger distortion parameter $\epsilon$ is allowed, the Type II error would decrease. This aligns with the intuition that adding statistical bias would make the output easier to detect (Aaronson, 2022a; Kirchenbauer et al., 2023a). Among all choices of $\epsilon$, the case $\epsilon = 0$ is of particular interest since it preserves the marginal distribution of the service provider's output. Therefore, we will focus on this distortion-free case in the following sections.

Recall that in practice, the watermarks are implemented via pseudo-random functions. Therefore, the uniformly most powerful test in Theorem C.2 is effectively using a pseudo-random generator to approximate the distribution $\rho$, combined with an $\alpha$-clipping to control Type I error. This construction reveals a surprising message: simply using pseudo-random generator to approximate the distribution is optimal. We illustrate the practical implementation of the UMP watermark in Algorithm 1 and 2.

**Remark C.3** (Watermarking guarantees). *To achieve the upper bound of Theorem C.2, the detector needs to access the model and the prompt in order to generate the reject region, which is not always*

*accessible in many real-world applications. Therefore, the upper bound of Theorem C.2 achieves a weaker watermarking guarantee compared with previous works (Aaronson, 2022a; Kirchenbauer et al., 2023a; Christ et al., 2023). In Appendix C.2, we study model-agnostic watermarking that overcomes this limitation.*

*Nonetheless, the lower bound in Theorem C.2 characterizes a fundamental limit of Problem B.1, thus providing an information-theoretic lower bound for all watermarks.*

**Remark C.4** (Use cases of the UMP watermark). *The utilization of the UMP watermark offers an efficient approach for LLM service providers to determine if instruction-following datasets have been generated by a specific model. In the context of instruction-following datasets, both the prompt and response are explicitly provided to the detectors, enabling the UMP watermark to perform accurate watermarking and detection without extra source of information. This usage is beneficial in identifying and filtering out data points that have been comtaminated by texts generated from models like GPT-4 (OpenAI, 2023), thereby preserving the purity and quality of the training data.*

### C.2 RATES OF MODEL-AGNOSTIC WATERMARKING

It is noticeable that for large $\mathcal{Q}$, a $\mathcal{Q}$-watermarking scheme can not perform as good as a watermarking specifically designed for $\rho$ for any distribution $\rho \in \mathcal{Q}$. This means that Uniformly Most Powerful $\mathcal{Q}$-Watermarking might not exist in general. To evaluate model-agnostic watermarking schemes, a natural desideratum is therefore the maximum difference between its Type II error and the Type II error of the UMP watermarking of $\rho$ over all distributions $\rho$, under fixed Type I error. Specifically, we introduce the following notion.

**Definition C.5** (Minimax most powerful model-agnostic watermark). We say that a $\mathcal{Q}$-agnostic watermark $(\eta, \{\mathcal{P}_\rho\}_{\rho \in \mathcal{Q}})$ is of level-$\alpha$ if the Type I error of $\mathcal{P}_\rho$ is less than or equal to $\alpha$ for any $\rho \in \mathcal{Q}$. Define the *maximum Type II error loss* of $(\eta, \{\mathcal{P}_\rho\}_{\rho \in \mathcal{Q}})$ as

$$\gamma(\eta) := \sup_{\rho \in \mathcal{Q}} \beta(\mathcal{P}_\rho) - \beta(\mathcal{P}_\rho^*)$$

where $\mathcal{P}_\rho^*$ is the UMP distortion-free watermark of $\rho$ of level $\alpha$.

We say that a $\mathcal{Q}$-agnostic watermarking scheme is minimax most powerful, if it minimizes the maximum Type II error loss among all $\mathcal{Q}$-agnostic watermarks of level $\alpha$.

The following result characterizes the Type II error loss of the minimax most powerful model-agnostic watermarking.

**Theorem C.6.** *Let $|\Omega| = m$ and suppose $\alpha m, \frac{1}{\alpha} \in \mathbb{Z}$.[2] In the minimax most powerful model-agnostic watermarking scheme of level-$\alpha$, the marginal distribution of the reject region is given by*

$$\eta^*(A) = \begin{cases} \frac{1}{\binom{m}{\alpha m}}, & \text{if } |A| = \alpha m \\ 0, & \text{otherwise} \end{cases}.$$

*The maximum Type II error loss of the minimax most powerful model-agnostic watermarking scheme of level-$\alpha$ is given by $\gamma(\eta^*) = \frac{\binom{m - \frac{1}{\alpha}}{\alpha m}}{\binom{m}{\alpha m}}$. In the regime $\alpha \to 0_+, m \to +\infty$, we have $\gamma(\eta^*) \to c$ for some constant $c \le e^{-1}$, and when $1/(\alpha m) \to 0_+$ is further satisfied, $c = e^{-1}$.*

**Remark C.7.** *The $e^{-1}$ maximum Type II error loss does not contradict with the $h^{-2}$ rates in previous works (Aaronson, 2022a; Christ et al., 2023; Kuditipudi et al., 2023), because as $n \gtrsim h^{-2}$, the model distribution over the sequences of $n$ tokens (with average entropy $h$ per token) is beyond the worst case. Indeed, such distributions have higher differential entropy than the hard instances in the proof.*

Remark C.7 highlights that the hard instance constructed in Theorem C.6 may possess a lower entropy than that of the actual model. Therefore, it raises an important question: for a smaller class $\mathcal{Q}$ that contains distributions with higher entropy, what is the minimum achievable Type II error loss for $\mathcal{Q}$-agnostic watermarking? It is obvious that the minimax rate over a higher entropy level should improve upon the previous rate of $e^{-1}$.

---

[2]For the general case, it suffices to let $a_1 = 1/(\lceil 1/\alpha \rceil)$ and $m_1 = \lceil \alpha_1 m \rceil / \alpha_1$ and augment $\Omega$ with $m_1 - m$ dummy outcomes. Then $\alpha_1 m, 1/\alpha_1 \in \mathbb{Z}$ and hence the minimax bound for the new sample space with cardinality $m_1$ and the new Type I error $\alpha_1$ yields a nearly-matching bound for $(m, \alpha)$.

Towards answering this question, we consider the following class of distributions:

$$\mathcal{Q}_\kappa := \left\{ \rho : \sup_{\omega \in \Omega} \rho(\{\omega\}) \le \kappa \right\}$$

where $\kappa$ represents the level of randomness and decreases as entropy increases. The *maximum Type II error loss* of $\mathcal{Q}_\kappa$-agnostic watermarking $(\eta, \{\mathcal{P}_\rho\}_{\rho \in \mathcal{Q}_\kappa})$ is thus given by

$$\gamma(\eta, \kappa) := \max_{\rho \in \Delta(\Omega) : \sup_{\omega \in \Omega} \rho(\{\omega\}) \le \kappa} \beta(\mathcal{P}_\rho) - \beta(\mathcal{P}_\rho^*)$$

where $\mathcal{P}_\rho^*$ is the UMP distortion-free watermark of $\rho$. The following result gives an upper bound of the above quantity, thus answering the question.

**Theorem C.8.** *Let $|\Omega| = m$ and suppose $\alpha m, \frac{1}{\kappa} \in \mathbb{Z}$. Then the maximum Type II error loss of the minimax $\mathcal{Q}_\kappa$-agnostic watermarking of level-$\alpha$ is given by*

$$\gamma(\eta^*, \kappa) = \frac{\binom{m - \alpha m}{1/\kappa}}{\binom{m}{1/\kappa}}.$$

The proof can be found in Appendix G.4. When $\kappa \le \alpha$, the Type II error of model-nonagnostic watermark vanishes, and therefore Theorem C.8 provides an upper bound $\frac{\binom{m - \alpha m}{1/\kappa}}{\binom{m}{1/\kappa}}$ for the worst-case Type II error. This rate improves over $e^{-1}$ in Theorem C.6. In the next section, we will apply Theorem C.8 to the i.i.d. setting where $\kappa$ can be exponentially small. This will lead to an negligible maximum Type II error loss for model-agnostic watermarking.

## C.3 RATES IN THE I.I.D. SETTING

In practice, the sample space $\Omega$ is usually a Cartesian product in the form of $\Omega_0^{\otimes n}$. For example, the output of LLMs takes form of a sequence of tokens, each coming from the same vocabulary set $V$. The quantity of practical interest becomes the minimum number of tokens to achieve certain statistical watermarking guarantees. In order to find the explicit rates on the minimum number of required tokens, we study the theoretical setting where the tokens are samples i.i.d. from a fixed distribution.

In this section, we consider the product distribution $\rho = \rho_0^{\otimes n}$ over $\Omega_0^{\otimes n}$ and the important setting of $\epsilon = 0$ (distortion-free watermarking). We introduce the following two quantities:

- Let $h$ denote the entropy of $\rho_0$. We use $n_{\mathrm{ump}}(h, \alpha, \beta)$ to denote the minimum number of tokens required by the UMP watermark to achieve Type I error $\le \alpha$ and Type II error $\le \beta$.
- Define $n_{\mathrm{minimax}}(h, \alpha, \beta)$ as the minimum number of tokens required by minimax $\mathcal{Q}^h$-agnostic watermark to achieve Type I error $\le \alpha$ and Type II error $\le \beta$, where $\mathcal{Q}^h := \{\rho = \rho_0^{\otimes n} : H(\rho_0) \ge h\}$, i.e. contains all distributions $\rho = \rho_0^{\otimes n}$ such that the entropy of $\rho_0$ is $\ge h$.

Together, $n_{\mathrm{ump}}(h, \alpha, \beta)$ and $n_{\mathrm{minimax}}(h, \alpha, \beta)$ serve as critical thresholds beyond which the desired statistical conclusions can be drawn regarding the output, making them essential parameters in watermarking applications.

We start by inspecting the rates in Theorem C.2 in the i.i.d. setting. The following result gives a nearly-matching upper bound and lower bound of $n_{\mathrm{ump}}(h, \alpha, \beta)$.

**Theorem C.9.** *Suppose $\alpha, \beta < 0.1$. We have $n_{\mathrm{ump}}(h, \alpha, \beta) \ge \Omega\left( \left( \frac{\ln \frac{1}{h} \left( \ln \frac{1}{\alpha} \wedge \ln \frac{1}{\beta} \right)}{h} \right) \vee \frac{\ln \frac{1}{\alpha}}{h} \right).$*

*Furthermore, let $k = |\Omega_0|$, we have $n_{\mathrm{ump}}(h, \alpha, \beta) \le O\left( \left( \frac{\ln \frac{k}{h} \cdot \left( \ln \frac{1}{\alpha} \wedge \ln \frac{1}{\beta} \right)}{h} \right) \vee \frac{\ln \frac{1}{\alpha} \ln k}{h} \right).$*

**Remark C.10** (Tightness). *Up to a constant and logarithmic factor in $k$, our upper bound matches the lower bound. Notice that since any model with an arbitrary token set can be reduced into a model with a binary token set (Christ et al., 2023) (i.e. $k = 2$), our bound is therefore tight up to a constant factor.*

Using Theorem C.8 and Theorem C.9, we are now in the position to characterize $n_{\mathrm{minimax}}(h, \alpha, \beta)$. Suppose the sample space is a Cartesian product $\Omega = \Omega_0^{\otimes n_0}$ and constrain to product measures over

sequences of $n_0$ tokens, like in Appendix C.3. We start by the following relationship:[3]

$$1 - \max_{\rho_0 : H(\rho_0) \geq h} \max_{\omega \in \Omega_0} \rho_0(\{\omega\}) \geq \Omega\left(\frac{h}{\ln(1/h)}\right)$$

where a detailed derivation can be found in Lemma G.3. It follows that

$$\kappa \leq \left(\max_{\rho_0 : H(\rho_0) \geq h} \max_{\omega \in \Omega_0} \rho_0(\{\omega\})\right)^{n_0} = e^{-\Omega\left(\frac{n_0 h}{\ln(1/h)}\right)}.$$

Using this observation and the derivation in Theorem C.6, $\gamma(\eta^*, \kappa)$ can be bounded by

$$(1-\alpha)^{1/\kappa} \leq (1-\alpha)^{e^{\Omega\left(\frac{n_0 h}{\ln(1/h)}\right)}}.$$

This means that when $n_0 \gtrsim \frac{\ln(1/h)}{h} \cdot (\ln(1/\alpha) + \ln\ln(1/\beta))$, the maximum Type II error loss given by Theorem C.8 and the Type II error of the UMP watermarking given in Theorem C.9 can be simultaneously bounded by $\beta$, thus establishing an upper bound. Furthermore, this rate matches the lower bound in Theorem C.9, where the guarantee is weaker (model-nonagnostic). Combining the above arguments, we obtain an nearly-matching upper and lower bound.

**Corollary C.11.** *Suppose $\alpha, \beta < 0.1$. We have*

$$n_{\mathrm{minimax}}(h, \alpha, \beta) = O\left(\frac{\ln(1/h)}{h} \cdot (\ln(1/\alpha) + \ln\ln(1/\beta))\right),$$

$$n_{\mathrm{minimax}}(h, \alpha, \beta) = \Omega\left(\frac{\ln(1/h)}{h} \cdot (\ln(1/\alpha) \wedge \ln(1/\beta))\right).$$

**Remark C.12** (Comparison with previous works). *As commented in Appendix C.1, the regime $h \ll 1$ is more important and challenging because it is when watermarking is challenging. In this regime, our rate of $\frac{\ln(1/h)}{h}$ improves the previous rate of $h^{-2}$ in a line of works (Aaronson, 2022a; Kirchenbauer et al., 2023a; Zhao et al., 2023; Liu et al., 2023; Kuditipudi et al., 2023), and highlights a fundamental gap between the existing watermarks and the information-theoretic lower bound.*

### C.4 RATES IN NON-IID SETTING

Without the i.i.d. condition, determining the explicit rates for the minimum number of tokens required to achieve fixed Type I and Type II errors is generally intractable. Indeed, with arbitrary token distributions, the probability of generating outputs with low empirical entropy might be $\Omega(1)$. Outputs with low empirical entropy are typically challenging to watermark (see, e.g., Appendix C.1 or Aaronson (2022b)), leading to a high rate of false negatives. Consequently, the Type II error may still be $\Omega(1)$ even when the number of tokens is sufficiently large. The next example formalizes this failure case.

**Example C.13** (High entropy, infinite tokens, still cannot watermark). Let the token space be $\{0, 1, \ldots, K\}$ and the auto-regressive distribution be given by

$$\mathbb{P}(X_1 = 0) = C > 0.5,$$

$$\mathbb{P}(X_1 = j) = \frac{1-C}{K}, \ \forall j \neq 0$$

$$\mathbb{P}(X_{i+1} = 0 | X_1 = \cdots = X_i = 0) = 1, \ \forall i \geq 1$$

$$\mathbb{P}(X_{i+1} = j | X_{1:i} = x_{1:i}) = \frac{1}{K}, \ \forall x_{1:i} \neq 0 \cdots 0, j \neq 0.$$

Then it can be verified that $\forall n \in \mathbb{Z}_+$:

$$\mathbb{P}(X_{1:n} = 0 \cdots 0) = C,$$

$$H(X_{1:n}) \geq n \cdot (1-C) \cdot \log \frac{K}{1-C}.$$

This indicates that the *average entropy per token is at least $(1-C) \cdot \log \frac{K}{1-C}$, which is high*. However, for any $n \in \mathbb{Z}_+$, *the sequence $X_{1:n}$ fails to be detected with probability at least $C > 0.5$, since the sequence $0 \cdots 0$ is nearly deterministic. Therefore, the Type II error is always $\Omega(1)$ no matter how many tokens are used, despite the average entropy per token being high.*

---

[3]In the rest of this section, we omit logarithmic factors in the cardinality of the vocabulary.

In this section, we present a different, 'conditional' guarantee overcome this limitation. This result bounds the conditional probability of false negatives among all outputs with high empirical entropy, while excluding outputs with low empirical entropy from consideration.

**Theorem C.14.** *For any $\alpha, \beta \in (0, 1)$, if*

$$n, \widehat{H} \gtrsim \log \frac{1}{\alpha} + \log \log \frac{1}{\beta}$$

*then for any distribution distribution $\rho$ over $\Omega = \Omega_0^{\otimes n}$, there exists a model-agnostic watermarking of level $\alpha$ such that the conditional probability that an output $X$ from the watermarked model is not rejected, given that the empirical entropy of $X$ is at least $\widehat{H}$, is less than or equal to $\beta$. More precisely, there exists a coupling $\mathcal{P}$ of $\rho$ and the $\eta^*$ defined in Theorem 3.6, such that*

$$\mathbb{P}_{(X,R)\sim\mathcal{P}}(X \notin R | - \log \rho(X) \geq \widehat{H}) \leq \beta$$
$$\sup_{\pi \in \Delta(\Omega)} \mathbb{P}_{Y\sim\pi, (X,R)\sim\mathcal{P}}(Y \in R) \leq \alpha.$$

The proof is deferred to Appendix G.5. Intuitively, Theorem C.14 suggests that long texts with high empirical entropy will be watermarked with high probability. It is important to note that $\widehat{H}$ is flexible, allowing us to bound the conditional Type II error for any class of outputs with empirical entropy levels above $\log \frac{1}{\alpha} + \log \log \frac{1}{\beta}$.

## C.5 EFFICIENCY-ROBUSTNESS TRADE-OFFS

In the context of watermarking large language models, it's crucial to acknowledge users' capability to modify or manipulate model outputs. These modifications include cropping, paraphrasing, and translating the text, all of which may be employed to subvert watermark detection. Therefore, in this section, we introduce a graphical framework, modified from Problem B.1, to account for potential user perturbations and investigate the optimal watermarking schemes robust to these perturbations. The formulation here shares similarity with a concurrent work by Zhang et al. (2023).

**Definition C.15** (Perturbation graph). A perturbation graph over the discrete sample space $\Omega$ is a directed graph $G = (V, E)$ where $V$ equals $\Omega$ and $(u, u) \in E$ for any $u \in V$. For any $v \in V$, let $in(v) = \{w \in V : (w, v) \in E\}$ denote the set of vertices with incoming edges to $v$, and let $out(v) = \{w \in V : (v, w) \in E\}$ denote the set of vertices with outcoming edges from $v$.

The perturbation graph specifies all the possible perturbations that could be made by the user: any $u \in V$ can be perturbed into $v \in V$ if and only if $(u, v) \in E$, i.e., there exists a directed edge from $u$ to $v$.

**Example C.16.** Consider $\Omega = \Omega_0^{\otimes n}$. Let the user have the capacity to change no more than $c$ tokens, i.e., perturb any sequence of tokens $x = x_1 x_2 \cdots x_n$ to another sequence $y = y_1 y_2 \cdots y_n$ with Hamming distance less than or equal to $c$. Then the perturbation graph is given by $G = (V, E)$ where $V = \Omega^n$ and $E = \{(u, v) : u, v \in V, d(u, v) \leq c\}$ ($d$ is the Hamming distance, i.e., $d(x, y) = \sum_{i=1}^{n} \mathbb{1}(x_i \neq y_i)$).

**Problem C.17** (Robust watermarking scheme). A robust watermarking scheme with respect to a perturbation graph $G$ is a watermarking scheme except that its Type II error is defined as $\mathbb{E}_{X,R\sim\mathcal{P}} \left[ \max_{Y \in out(X)} \mathbb{1}(Y \notin R) \right]$, i.e., the probability of false negative given that the user adversarially perturbs the output.

The next result characterize the optimum Type II error achievable by robust watermarking, where the proof can be found in Appendix G.6.

**Theorem C.18.** *Define the shrinkage operator $\mathcal{S}_G : 2^\Omega \to 2^\Omega$ (of a perturbation graph $G$) by $\mathcal{S}_G(R) = \{x \in \Omega : out(x) \subset R\}$ and its inverse $\mathcal{S}_G^{-1}(R) = \cup_{x \in R} out(x)$. Then the minimum Type II error of the robust, 0-distorted UMP test of level $\alpha$ in Problem C.17 is given by the solution of the*

*following Linear Program*

$$\min_{x \in \mathbb{R}^{|\Omega|}} 1 - \sum_{y \in \Omega} \rho(y) x(y) \tag{2}$$

$$s.t. \sum_{y \in in(z)} \rho(y) x(y) \le \alpha, \sum_{z \in \Omega} x(z) \le 1,$$

$$0 \le x(z) \le 1, \ \forall z \in \Omega.$$

*The UMP watermarking is given by* $\mathcal{P}^*(X = y, R = R_0) =$
$$\begin{cases} \rho(y) \cdot x^*(y), & R_0 = \mathcal{S}_G^{-1}(\{y\}) \\ \rho(y) \cdot (1 - x^*(y)), & R_0 = \emptyset \\ 0, & otherwise \end{cases} \quad \text{where } x^* \text{ is the solution of Eq. (2).}$$

**Remark C.19** (Dependence on the sparsity of graph). *From Eq. (2), we observe that the perturbation graph influence the optimal Type II error via the constraint set. Indeed, if the graph is dense, the constraints $\sum_{y \in in(z)} \rho(y) x(y) \le \alpha$ involve many entries of $y \in \Omega$ and thus decrease the value $\sum_{y \in \Omega} \rho(y) x(y)$, thereby increasing the Type II error. On the other extreme, when the edge set of the perturbation graph is $E = \{(u, u) : u \in v\}$, i.e., the user can not perturb the output to a different value, then optimum of Eq. (2) reduces to the rates in Theorem C.2 (setting $\epsilon = 0$).*

## D ALGORITHM PSEUDOCODES

### D.1 UMP WATERMARK

In this section, we present the pseudocode of the UMP watermark, i.e., the watermarking scheme achieving the Type II error in Theorem 2.1. The watermark generation and detection pipelines are outlined in Algorithm 1-2 respectively. These pseudocodes are used to generate the results in Table 6.

In Table 3, we introduce the parameters used in our algorithm.

| Parameter | Meaning |
|:---:|:---:|
| $p$ | prompt |
| $sk$ | secret key |
| $\rho_{j+1}$ | next token distribution |
| $\mathcal{S}$ | EOS token set |
| $\tau$ | temperature of $M$ |
| $M$ | target (watermarked) language model |
| PRG | pseudo-random generator |
| $\alpha$ | Type I error |

Table 3: Parameters in UMP watermark (Theorem 2.1)

### D.2 SEAL

In this section, we describe the pseudocode of SEAL, shown Algorithm 3-4. We will use a random hash function that maps a token to a hash code. The main idea of our watermarking method is to couple a proposal model $M_P$ and the hash code (random seed), so that the watermarked tokens hashes to the random seed with high probabilities.

To sample $j$-th token in the generation phase, we use the proposal model $M_P$ and the (randomly selected) hashing function $h_j$ to sample a random seed $s_j$ from $h_j \sharp q_j$ (the pushforward of $q_j$ by

---

[3]$y$: the text to be detected.

---

**Algorithm 1** UMP Generation

---

1: **procedure** GENERATION($p, \tau, M(\cdot; \cdot), r$, PRG$(\cdot, \cdot)$, EOS)
2:                                                      ▷ Generate a watermarked sequence of tokens $t = t_1 \ldots t_j$
3:     $j \leftarrow 0$
4:     **while** True **do**
5:         $\rho_j \leftarrow M(p \parallel t_{1:j-1}, \tau)$                     ▷ Distribution of the $j$-th token according to $M$
6:         $t_j \leftarrow$ PRG$(\rho_j, sk)$
7:         **if** $t_j \notin$ EOS **then**
8:             $j \leftarrow j + 1$
9:         **else**
10:            break;
11:         **end if**
12:     **end while**
13:     **Return** $t_{1:j}$
14: **end procedure**

---

**Algorithm 2** UMP Detection

---

1: **procedure** DETECTION($y, p, \tau, M(\cdot, \cdot), \alpha, sk$, PRG$(\cdot, \cdot)$, EOS)
2:                                               ▷ Detect whether a given sequence $y$ is watermarked.
3:     $j \leftarrow 0, pv \leftarrow 1$                                          ▷ $pv$ is the p-value
4:     **while** True **do**
5:         $\rho_j \leftarrow M(p \parallel s_{1:j-1}, \tau)$                  ▷ Distribution of the $j$-th token according to $M$
6:         $s_j \leftarrow$ PRG$(\rho_j, sk)$
7:         **if** $s_j \notin$ EOS **then**
8:             $j \leftarrow j + 1, pv \leftarrow pv \times \rho_j(s_j)$
9:         **else**
10:             break;
11:         **end if**
12:     **end while**
13:     **if** $s = y$ and $pv \leq \alpha$ **then**
14:         `watermarked` = True                        ▷ Flag if the p-value is less than $\alpha$
15:     **else**
16:         `watermarked` = False
17:     **end if**
18:     **Return** `watermarked`
19: **end procedure**

---

$h_j$), where $q_j$ is the proposal model's conditional distribution of the next token on the previous $K$ tokens $t_{j-K:j-1}$. The sampling steps are implemented by a pseudo-random generator PRG using key $key$, where the key is computed using the last $H$ tokens and the secret key $sk$. In practice, one may also use external pseudo-randomness generator to set the secret keys (Kuditipudi et al., 2023; Piet et al., 2023). We add logits bias $\delta$ to the pre-image of $s_j$ under mapping $h$ via formula $\rho_j(t) \leftarrow \rho_j(t) \cdot e^{\mathbb{1}(h_j(t)=s_j) \cdot \delta/\tau}/Z, \ \forall t$, where $Z$ is the normalizing constant. Then with probability $\frac{h_j \sharp \rho_j(s_j)}{h_j \sharp q_j(s_j)} \wedge 1$, the proposal code $s_j$ is accepted to be true code $c_j$. Here, $h_j \sharp \rho_j(s_j)$ and $h_j \sharp q_j(s_j)$ denote the probability of $s_j$ under the pushforward of target model $M$ and proposal model $M_P$'s distributions, respectively. If $s_j$ is not accepted, then we sample the true code $c_j$ from the distribution $(h_j \sharp \rho_j - h_j \sharp q_j)_+$. Finally, we sample the next token from $\rho_j$ conditional on that the next token hashes to the true code $c_j$.

To detect whether a sequence of tokens $y_{1:n}$ is watermarked, we reproduce the random seeds $s_{K:n}$ using the same proposal model $M_P$, pseudo-random generator PRG, and secret key $sk$. In the ideal case where the sequence is not perturbed, the seeds $s_{K:n}$ should be exactly the same as those in the generation phase since the same key and pseudo-random function are used. Then, we check whether the token $y_j$ hashes to the code $s_{K:n}$, indicated by $\xi_j$. Finally, we may say a subsequence $y_{i:i+L}$ is watermarked if the following condition holds:

$$\sum_{j=i}^{i+L} \xi_j \geq 1 - \alpha \text{ quantile of the random variable } \left( \sum_{j=i}^{i+L} Z_j \right) \tag{3}$$

where $Z_j$ is Bernoulli random variable with expectation $h_j \sharp q_j(h_j(t_j))$, independently from each other. To find the $1 - \alpha$ quantile, the probability mass function of $\sum_{j=i}^{i+L} Z_j$ can be computed iteratively by dynamic programming:

$$p_{1,1} = w_i, \; p_{1,0} = 1 - w_i$$
$$p_{k+1,l} = w_{i+k} \cdot p_{k,l-1} + (1 - w_{i+k}) \cdot p_{k,l} \tag{4}$$

where $w_j = h_j \sharp q_j(h_j(t_j)) = \mathbb{P}(Z_j = 1)$ and $p_{k,-1} = 0, p_{k,l} = 0 \; (k < l)$ by default. Then it can be verified that $p_{k,l} = \mathbb{P}\left( \sum_{j=i}^{i+k-1} Z_j = l \right)$.

Alternatively, we may also use Bernoulli concentration inequalities to approximate the threshold:

$$\sum_{j=i}^{i+L} \xi_j \geq (1 + \epsilon) \cdot \sum_{j=i}^{i+L} h_j \sharp q_j(h_j(t_j)) \tag{5}$$

where $\epsilon$ satisfies

$$(\epsilon - (1 + \epsilon) \log(1 + \epsilon)) \cdot \sum_{j=i}^{i+L} h_j \sharp q_j(h_j(t_j)) \leq \log \alpha.$$

In Table 4, we introduce the parameters used in our algorithm.

| Parameter | Meaning |
|-----------|---------|
| $p$ | prompt |
| $sk$ | secret key |
| $K$ | attention window length |
| $H$ | hash window length |
| KEYGEN | embedded key generator |
| $\rho_{j+1}$ | next token distribution |
| $\mathcal{S}$ | EOS token set |
| $\tau$ | temperature of $M$ |
| $\tau_P$ | temperature of $M_P$ |
| $\delta$ | bias |
| $M_P$ | smaller language model |
| $M$ | target (watermarked) language model |
| PRG | pseudo-random generator |
| $\mathcal{H}$ | family of hashing function |
| $\alpha$ | Type I error |

Table 4: Parameters in SEAL

---

**Algorithm 3** SEAL Generation

---

1: **procedure** GENERATION($p, \tau, \tau_P, \delta, M(\cdot, \cdot), M_P(\cdot, \cdot), sk, \text{PRG}(\cdot, \cdot), K, H, \text{EOS}, \mathcal{H}$)
2:                                     $\triangleright$ Generate a watermarked sequence of tokens $t = t_1 \ldots t_j$
3:      $j \leftarrow 0$
4:      **while** True **do**
5:          $q_j \leftarrow M_P(t_{j-K:j-1}, \tau_P)$                       $\triangleright$ Distribution of the $j$-th token according to $M_P$
6:          $\rho_j \leftarrow M(p \parallel t_{1:j-1}, \tau)$                       $\triangleright$ Distribution of the $j$-th token according to $M$
7:          $key \leftarrow \text{KEYGEN}(t_{i-H:i-1}; sk)$                $\triangleright$ Compute key by embedded key generator
8:          Sample $h_j$ from $\mathcal{H}$ under $key$                  $\triangleright$ Select a random hash function
9:          $s_j \leftarrow \text{PRG}(h_j \sharp q_j, key)$             $\triangleright$ Sample a random seed from $h_j \sharp q_j$ under key $key$
10:        $\rho_j(t) \leftarrow \rho_j(t) \cdot e^{\mathbb{1}(h_j(t) = s_j) \cdot \delta / \tau} / Z, \ \forall t$                       $\triangleright$ Add bias
11:        Sample $u \sim \text{Unif}(0, 1)$
12:        **if** $u < \frac{h_j \sharp \rho_j(s_j)}{h_j \sharp q_j(s_j)}$ **then**                        $\triangleright$ Speculative decoding
13:           $c_j \leftarrow s_j$
14:        **else**
15:           Sample $c_j \propto (h_j \sharp \rho_j - h_j \sharp q_j)_+$;                $\triangleright$ Maintain distortion-free
16:        **end if**
17:        Sample $t_j$ from $\rho_j (\cdot | h(t_j) = c_j)$              $\triangleright$ Conditional on the hash code
18:        **if** $t_j \notin \text{EOS}$ **then**
19:           $j \leftarrow j + 1$
20:        **else**
21:           break;
22:        **end if**
23:      **end while**
24:      **Return** $t_{1:j}$
25: **end procedure**

---

**Algorithm 4** SEAL Detection

---

1: **procedure** DETECTION($y, p, \tau_P, M_P(\cdot, \cdot), \alpha, sk, \text{PRG}(\cdot, \cdot), \mathcal{H}, \text{EOS}, K, H$)
2:                                  $\triangleright$ Detect whether a given sequence $y$ is watermarked.
3:      $j \leftarrow 0$
4:      **while** $y_j \notin \text{EOS}$ **do**
5:          $q_j \leftarrow M_P(y_{j-K:j-1}, \tau_P)$                        $\triangleright$ Recompute the probability from $M_P$
6:          $key \leftarrow \text{KEYGEN}(t_{i-H:i-1}; sk)$               $\triangleright$ Compute key by embedded key generator
7:          Sample $h_j$ from $\mathcal{H}$ under $key$                  $\triangleright$ Select a random hash function
8:          $s_j \leftarrow \text{PRG}(h_j \sharp q_j, key)$                   $\triangleright$ Reconstruct the random seed
9:          $\xi_j \leftarrow \mathbb{1}(h_j(y_j) = s_j)$                  $\triangleright$ Indicator of if $y_j$ hashes to $s_j$
10:        $j \leftarrow j + 1$
11:      **end while**
12:      **if** Eq. (3) holds for some $K \leq i < i + L$ **then**
13:        `watermarked` = True for $y_{i:i+L}$            $\triangleright$ Detect if $y_{i:i+L}$ has lot of hash collisions
14:      **else**
15:        `watermarked` = False for $y_{i:i+L}$
16:      **end if**
17:      **Return** `watermarked`
18: **end procedure**

---

# E    Experiment Details

In Table 1 and 2, the experiments on baselines follow the same parameters recommended by Piet et al. (2023). Since Phi-3-mini-128k-instruct has 64 additional tokens compared to Llama2-7B-chat (mostly used to format the system/user prompts), we truncate the logits from Phi-3-mini-128k-instruct to match the logits from Llama2-7B-chat in size. In SEAL configuration, we use $H = 1, K = 15, |\Omega_h| = 5, \tau_P = 1, \delta = 4$ for $T = 0.3$; $H = 1, K = 10, |\Omega_h| = 4, \tau_P = 1, \delta = 4$ for $T = 0.7$; $H = 1, K = 20, |\Omega_h| = 4, \tau_P = 1.25, \delta = 2$ for $T = 1$. These hyper-parameters are consistent across Table 1 and 2. We didn't conduct experiments for SEAL under temperature 0, because SEAL can be reduced to Distribution Shift watermark (Kirchenbauer et al., 2023a) by setting $\tau_P = \infty$ and $h$ as randomly assign to green or red at this temperature.

# F    Supplemental Experiment Results

## F.1    Ablation Studies

We study the impact of proposal models on the performance of SEAL. With other parameters fixed, we consider three different proposal models with model sizes scaling from 1B to 7B: TinyLlama-1.1B-Chat-v1.0 (Zhang et al., 2024), Phi-3-mini-128k-instruct (Abdin et al., 2024), vicuna-7b-v1.5 (Zheng et al., 2023). From Table 5, we observe that the proposal model has only a marginal impact on SEAL's performance. We hypothesize that this is because in SEAL the proposal model only attends to the last K tokens. This fixed context window likely minimizes variations in next-token probability predictions across different proposal models.

| Temperature | Proposal | Quality ($\uparrow$) | Size ($\downarrow$) | Tamper-resistance ($\uparrow$) |
|---|---|---|---|---|
| | TinyLlama-1.1B | 0.89 | 63 | 1.0 |
| 0.3 | Phi-3-mini (3.8B) | 0.90 | 57.5 | 1.0 |
| | vicuna-7b-v1.5 | 0.89 | 61 | 1.0 |
| | TinyLlama-1.1B | 0.89 | 49 | 1.0 |
| 0.7 | Phi-3-mini (3.8B) | 0.88 | 45.5 | 1.0 |
| | vicuna-7b-v1.5 | 0.89 | 48 | 1.0 |
| | TinyLlama-1.1B | 0.87 | 76 | 1.0 |
| 1.0 | Phi-3-mini (3.8B) | 0.88 | 75 | 1.0 |
| | vicuna-7b-v1.5 | 0.87 | 72 | 1.0 |

Table 5: Comparison of SEAL with different proposal models across quality, size, and tamper-resistance at different temperature settings.

## F.2    Empirical Validation of Theoretical Results

| Scheme / Temperature | 0 | 0.3 | 0.7 | 1 |
|---|---|---|---|---|
| Distribution Shift (Kirchenbauer et al., 2023a) | **53** | **55** | 50 | 114.5 |
| Exponential (Aaronson, 2022b) | impossible | 980 | 178.5 | 90 |
| Inverse Transform (Kuditipudi et al., 2023) | impossible | $\infty$ | 457.5 | 195 |
| Binary (Christ et al., 2023) | impossible | $\infty$ | $\infty$ | 372.5 |
| Theorem 2.1 | impossible | 60.5 | **24** | **15** |

Table 6: Theorem 2.1 and previous works tested on MARKMYWORDS (Piet et al., 2023). For each watermark scheme and each temperature, we show the median number of tokens required to detect the watermark at a given p-value of $\alpha = 0.02$. For the first four rows, one can refer to Piet et al. (2023); $\infty$ means over half of all generations are not watermarked and "impossible" means when the temperature is 0, the text generation procedure is deterministic and the entropy is zero, and thus any distortion-free watermark scheme does not work.

We show experimental results of the statistically-optimal watermark indicated by Theorem 2.1 along with several previous works, in term of the median number of tokens needed to detect the watermark.

We use Algorithm 1-2 to simulate Theorem 2.1. Table 6 shows that Theorem 2.1 needs significantly fewer tokens to detect the watermark in high temperature, which echos with Theorem 2.1's statistical optimality. An exception is that for the distribution shift scheme (Kirchenbauer et al., 2023b) with low temperature (0.3 and 0.1), where the number of tokens required by the distribution shift scheme is smaller. This is because distribution shift is not distortion-free while Theorem 2.1 only characterizes the limits of unbiased (distortion-free) watermarking. Note that Theorem 2.1 is experimented under the model non-agnostic setting (but its rate in the model-nonagnostic setting is not fundamentally different from that in the model-agnostic setting, due to Theorem C.8 and Corollary C.11) without considering robustness, while the four baseline schemes also work for model agnostic setting with robustness guarantees. Noting that the optimal algorithm implied by Theorem 2.1 is equivalent to the Log Likelihood Ratio test (Hu et al., 2023; Christ et al., 2023; Li et al., 2024), we have yet to see any other model-nonagnostic watermark in the literature. Therefore, our experiments is used only to exhibit the statistical limits in Type I&II errors and highlight the fundamental gap, instead of advocating for the superiority of any particular watermarking scheme.

# G    PROOFS

## G.1    PROOF OF THEOREM C.2

*Proof.* Let $\rho'$ denote the marginal probability of $X$ and let $\eta$ denote the marginal probability of $R$. In the bound of Type I error, choosing $\pi = \delta_y$ yields

$$
\begin{aligned}
\alpha &\geq \mathbb{P}_{X \sim \pi, R \sim \mathcal{P}(\Omega, \cdot)}(X \in R) \\
&= \mathbb{P}_{R \sim \eta}(y \in R) \\
&= \sum_{R \in 2^\Omega} \left( \sum_{x \in \Omega} \rho'(x)\mathcal{P}(R|x) \right) \cdot \mathbb{1}(y \in R).
\end{aligned} \tag{6}
$$

Now notice that

$$
\begin{aligned}
\mathcal{P}(X \in R) &= \mathbb{E}_{\mathcal{P}}[\mathbb{1}(X \in R)] \\
&= \sum_{y \in \Omega} \sum_{R \in 2^\Omega} \rho'(y)\mathcal{P}(R|y)\mathbb{1}(y \in R) \\
&= \sum_{y \in \Omega} \underbrace{\left( \sum_{R \in 2^\Omega} \rho'(y)\mathcal{P}(R|y) \cdot \mathbb{1}(y \in R) \right)}_{A(y)}.
\end{aligned}
$$

For the term $A(y)$, we first know that $A(y) \leq \rho'(y)$. Applying Eq. (6), we further have

$$
\begin{aligned}
A(y) &\leq \sum_{R \in 2^\Omega} \left( \sum_{x \in \Omega} \rho'(x)\mathcal{P}(R|x) \right) \cdot \mathbb{1}(y \in R) \\
&\leq \alpha.
\end{aligned}
$$

Combining the above two inequalies, it follows that

$$
\begin{aligned}
\mathcal{P}(X \in R) &\leq \sum_{y \in \Omega} (\alpha \wedge \rho'(y)) \\
&= 1 - \sum_{x \in \Omega: \rho'(x) > \alpha} (\rho'(x) - \alpha) \\
&\leq 1 - \min_{\mathrm{TV}(\rho' \| \rho) \leq \epsilon} \sum_{x \in \Omega: \rho'(x) > \alpha} (\rho'(x) - \alpha) \\
&\leq 1 - \left( \sum_{x \in \Omega: \rho(x) > \alpha} (\rho(x) - \alpha) - \epsilon \right)_+
\end{aligned}
$$

where first equality is achieved by

$$\rho' = \arg \min_{\mathrm{TV}(\rho'\|\rho) \leq \epsilon} \sum_{x \in \Omega : \rho'(x) > \alpha} (\rho'(x) - \alpha)$$

and the third inequality is achieved when $\sum_{x \in \Omega : \rho(x) < \alpha} (\alpha - \rho(x)) \geq \epsilon$, a sufficient condition for which being $|\Omega| \geq (2 + \epsilon)/\alpha$ (indeed, otherwise we have $1 \geq \sum_{x : \rho(x) < \alpha} \rho(x) > \alpha \cdot |\{x : \rho(x) < \alpha\}| - \epsilon \geq \alpha \cdot (|\Omega| - 1/\alpha) - \epsilon = 1$, a contradiction). This establishes the optimal Type II error.

Finally, to verify that $\mathcal{P}^*$ satisfies the conditions, the condition $\mathrm{TV}(\mathcal{P}^*(\cdot, 2^\Omega)\|\rho) \leq \epsilon$ is apparently satisfied. For any $y \in \Omega$ we have

$$\mathbb{P}_{R \sim \eta}(y \in R) = \sum_{x \in \Omega} \rho^*(x) \cdot \mathbb{P}(R = \{x\}) \cdot \mathbb{1}(y = x)$$

$$= \rho^*(y) \cdot \left(1 \wedge \frac{\alpha}{\rho^*(y)}\right)$$

$$\leq \alpha.$$

This implies the $\sup_{\pi \in \Delta(\Omega)} \mathbb{P}_{Y \sim \pi, (X,R) \sim \mathcal{P}^*}(Y \in R) \leq \alpha$ because any $\pi$ can be written as linear combination of $\delta_y$. Moreover,

$$\mathcal{P}^*(X \in R) = \sum_{x \in \Omega} \rho^*(x) \cdot \mathbb{P}(R = \{x\})$$

$$= \sum_{y \in \Omega} (\alpha \wedge \rho^*(y))$$

$$= 1 - \sum_{x \in \Omega : \rho^*(x) > \alpha} (\rho^*(x) - \alpha).$$

This verifies that $\rho^*$ achieves the advertised Type II error. $\qquad\square$

### G.2 PROOF OF THEOREM C.9

*Proof.* Throughout the proof we assume that $h < 1/4$, otherwise the bounds become trivial.

We first prove the lower bound. For this purpose, we construct the hard instance: let $q_0 = H_b^{-1}(h)$ (take the one $\geq 1/2$) where $H_b$ is the binary entropy function defined by $H_b(x) = -x \ln x - (1 - x) \ln(1 - x)$, and set $\rho_0 = (1 - q_0)\delta_{x_1} + q_0 \delta_{x_2}$ where $x_1, x_2$ are two different elements in $\Omega_0$. Then Lemma G.2 implies that $q_0 \geq 3/4$. By Theorem C.2,

$$\beta = 1 - \mathcal{P}(X \in R) = \sum_{x \in \Omega : \rho(x) > \alpha} (\rho(x) - \alpha)$$

$$\geq \frac{1}{2} \cdot \mathbb{P}\left(\rho(X) \geq 2\alpha\right)$$

$$= \frac{1}{2} \cdot \mathbb{P}\left(\sum_{i=1}^n \ln \rho_0(X_i) \geq \ln(2\alpha)\right)$$

$$\geq \mathbb{1}(n \ln q_0 \geq \ln(2\alpha)) \cdot \frac{1}{2} q_0^n$$

$$\geq \mathbb{1}(2n(1 - q_0) \leq -\ln(2\alpha)) \cdot \frac{1}{2} \exp\left(-2n(1 - q_0)\right)$$

$$\geq \mathbb{1}\left(n \leq \frac{\ln \frac{1}{2\alpha}}{2h/\ln \frac{\ln 2}{h}}\right) \cdot \frac{1}{2} \exp\left(-\frac{2nh}{\ln \frac{\ln 2}{h}}\right)$$

where the last inequality follows from Lemma G.3. It follows that

$$n(h, \alpha, \beta) \geq \frac{\ln \frac{\ln 2}{h}}{2h} \cdot \left(\ln \frac{1}{2\alpha} \wedge \ln \frac{1}{2\beta}\right). \tag{7}$$

Furthermore, suppose $n \leq \frac{\ln \frac{1}{2\alpha}}{4(1-q_0) \ln \frac{1}{1-q_0}}$. Define $Y = \sum_{i=1}^n \mathbb{1}(\rho_0(X_i) = 1 - q_0)$, then notice that $Y \sim \mathrm{Binom}(n, 1 - q_0)$ and if $Y \leq \frac{\ln \frac{1}{2\alpha}}{2 \ln \frac{1}{1-q_0}}$, then

$$\sum_{i=1}^n \ln \rho_0(X_i) \geq \frac{\ln \frac{1}{2\alpha}}{2 \ln \frac{1}{1-q_0}} \cdot \ln(1 - q_0) + n \cdot \ln q_0$$

$$\geq \ln(2\alpha)$$

where the last inequality is due to $n \cdot \ln q_0 \geq -2(1-q_0)n \geq -2(1-q_0) \frac{\ln \frac{1}{2\alpha}}{4(1-q_0) \ln \frac{1}{1-q_0}} = \frac{\ln(2\alpha)}{2 \ln \frac{1}{1-q_0}} \geq \frac{\ln(2\alpha)}{2}$. Applying this and Markov's inequality,

$$\mathbb{P}\left(\sum_{i=1}^n \ln \rho_0(X_i) \geq \ln(2\alpha)\right) \geq \mathbb{P}\left(Y \leq \frac{2 \ln \frac{1}{2\alpha}}{\ln \frac{1}{1-q_0}}\right)$$

$$\geq 1 - \frac{n(1-q_0)}{\frac{2 \ln \frac{1}{2\alpha}}{\ln \frac{1}{1-q_0}}}$$

$$\geq \frac{1}{2},$$

A contradiction to $\mathbb{P}(\rho(X) \geq 2\alpha) \leq 2\beta$. As a result,

$$n(h, \alpha, \beta) \geq \frac{\ln \frac{1}{2\alpha}}{4(1-q_0) \ln \frac{1}{1-q_0}}$$

$$\geq \frac{\ln \frac{1}{2\alpha}}{4h}. \tag{8}$$

Combining Eq. (7) and Eq. (8), we established the lower bound.

For the upper bound, we define $q = \max_{x \in \Omega_0} \rho_0(x)$, then Lemma G.2 implies that $q \geq 1/2$. Define $Y = \sum_{i=1}^n \mathbb{1}(\rho_0(X_i) \neq q)$ (recall that $Y \sim \mathrm{Binom}(n, 1-q)$). It suffices to show when

$$n = 900 \left(\frac{2 \ln \frac{9k}{h}}{h} \cdot \left(\ln \frac{1}{\alpha} \wedge \ln \frac{1}{\beta}\right)\right) \vee \frac{(18 + 36 \ln(9k)) \ln \frac{1}{\alpha}}{h}$$

the Type II error of the UMP watermark $1 - \mathcal{P}^*(X \in R) \leq \beta$.

By Theorem C.2 and Bennett's inequality,

$$1 - \mathcal{P}^*(X \in \mathbb{R}) = \sum_{x \in \Omega : \rho(x) > \alpha} (\rho(x) - \alpha)$$

$$\leq \mathbb{P}(\rho(X) \geq \alpha)$$

$$= \mathbb{P}\left(\sum_{i=1}^n \ln \rho_0(X_i) \geq \ln(\alpha)\right)$$

$$\leq \mathbb{P}\left(Y \leq \frac{\ln \frac{1}{\alpha}}{\ln \frac{1}{1-q}}\right)$$

$$\leq \exp\left(-nq(1-q)\theta\left(\frac{1 - q - \frac{\ln \frac{1}{\alpha}}{n \ln \frac{1}{1-q}}}{q(1-q)}\right)\right) \tag{9}$$

where $\theta(x) = (1+x) \ln(1+x) - x$; the penultimate inequality follows from $\sum_{i=1}^n \ln \rho_0(X_i) \leq Y \ln(1-q)$.

Notice that by Lemma G.2,

$$(1-q)\ln\frac{1}{1-q} \geq \frac{h}{9\ln\frac{9k\ln(9k)}{h}} \cdot \ln\frac{\ln\frac{1}{h}}{h}$$

$$= h \cdot \frac{\ln\ln\frac{1}{h} + \ln\frac{1}{h}}{9\left(\ln\frac{1}{h} + \ln(9k\ln(9k))\right)}$$

$$\geq \frac{h}{9 + \ln(9k\ln(9k))}.$$

Since $n \geq \frac{(18 + 36\ln(9k\ln(9k)))\ln\frac{1}{\alpha}}{h}$, we have $n \geq \frac{2}{1-q}\frac{\ln\frac{1}{\alpha}}{\ln\frac{1}{1-q}}$. Under this condition, we have the simplification

$$\theta\left(\frac{1-q-\frac{\ln\frac{1}{\alpha}}{n\ln\frac{1}{1-q}}}{q(1-q)}\right) \geq \theta\left(\frac{1}{2q}\right)$$

$$\geq \frac{1}{50}.$$

Plugging back to Eq. (9), we have

$$1 - \mathcal{P}^*(X \in \mathbb{R}) \leq \exp\left(-nq(1-q)\theta\left(\frac{1-q-\frac{\ln\frac{1}{\alpha}}{n\ln\frac{1}{1-q}}}{q(1-q)}\right)\right)$$

$$\leq \exp\left(-\frac{n(1-q)}{100}\right)$$

$$\leq \exp\left(-\frac{nh}{900\ln\frac{9k\ln(9k)}{h}}\right) \tag{10}$$

where we applied Lemma G.3 in the last step.

Furthermore, we have

$$1 - \mathcal{P}(X \in R) \leq \mathbb{P}\left(\sum_{i=1}^{n}\ln\rho_0(X_i) \geq \ln(\alpha)\right)$$

$$\leq \mathbb{1}\left(n \leq \frac{\ln\alpha}{\ln q}\right)$$

$$\leq \mathbb{1}\left(n \leq 900\left(\frac{\ln\frac{9k\ln(9k)}{h}}{h} \cdot \ln\frac{1}{\alpha}\right)\right) \tag{11}$$

where the last step is due to Lemma G.3. Combining Eq. (10) and Eq. (11), we know that $1 - \mathcal{P}^*(X \in \mathbb{R}) \leq \beta$ when $n \geq 900\left(\frac{\ln\frac{9k\ln(9k)}{h}}{h} \cdot \left(\ln\frac{1}{\alpha} \wedge \ln\frac{1}{\beta}\right)\right)$. This establishes the upper bound. $\qquad\square$

### G.2.1 Supporting lemmata

**Lemma G.1** (Topsøe (2001), Theorem 1.2). *Define the binary entropy function $H_b : (0,1) \to \mathbb{R}$ as $H_b(x) = -x\ln x - (1-x)\ln(1-x)$. Then $4x(1-x) \leq H_b(x) \leq (4x(1-x))^{1/\ln 4}$.*

**Lemma G.2.** *Suppose $\rho$ is a probability measure over $\Omega$ such that $H(\rho) = h$, define $q = \max_{x\in\Omega}\rho(x)$. If $H(\rho) \leq 1/4$, then $q \geq 1/2$. Furthermore, if $H_b(q) \leq 1/4$, then $q \geq 3/4$.*

*Proof.* Suppose $q \leq 1/2$. By convexity of $H$,

$$H(\rho) \geq -\left\lfloor\frac{1}{q}\right\rfloor q\ln q \geq -\frac{1}{2}\ln\frac{1}{2} \geq 1/4.$$

This is a contradiction.

Suppose $q \leq 3/4$, then Lemma G.1 implies that

$$H_b(q) \geq 4q(1-q) \geq 1/4.$$

This is a contradiction. □

**Lemma G.3.** *Suppose $\rho$ is a probability measure over $\Omega$ such that $H(\rho) = h$ and $|\Omega| = k$. Define $q = \max_{x \in \Omega} \rho(x)$. If $q \geq 1/2$, then we have*

$$\frac{h}{9 \ln \frac{9k \ln(9k)}{h}} \leq 1 - q \leq \frac{h}{\ln \frac{\ln 2}{h}}$$

*Proof.* We have

$$H(\rho) \geq -(1-q)\ln(1-q) \geq (1-q) \cdot \ln 2.$$

It follows that

$$h \geq -(1-q)\ln(1-q)$$

$$\geq (1-q)\ln\frac{\ln 2}{h}.$$

Therefore $1 - q \leq \frac{h}{\ln \frac{\ln 2}{h}}$.

By the convexity of $H$ and $-q \ln q \leq 2(1-q)$,

$$H(\rho) \leq -q\ln q - (1-q)\ln\frac{1-q}{k}$$

$$\leq (1-q)\ln\frac{9k}{1-q}.$$

This means that

$$h^2 \leq (1-q)^2 \left(\ln\frac{9k}{1-q}\right)^2$$

$$\leq 2(1-q)^2 \left(\ln^2(9k) + \ln^2(1-q)\right)$$

$$\leq 2(1-q)^2 \ln^2(9k) + (1-q) \cdot \left(2(1-q)\ln^2(1-q)\right)$$

$$\leq (1-q) \cdot (\ln^2(9k) + 18) \tag{12}$$

where the last inequality is due to $2(1-q) \leq 1$ and $2(1-q)\ln^2(1-q) \leq 18$. It follows that

$$h \leq (1-q)\ln\frac{9k}{1-q}$$

$$\leq 9(1-q)\ln\frac{9k\ln(9k)}{h}$$

where the last step is because $\ln\frac{1}{1-q} \leq 2\ln\frac{\ln^2(9k)+18}{h} \leq 9\ln\frac{\ln(9k)}{h}$, using Eq. (12). This establishes $1 - q \geq \frac{h}{9 \ln \frac{9k\ln(9k)}{h}}$. □

## G.3 PROOF OF THEOREM C.6

*Proof.* **Lower bound.** By definition of Type I error, for any level-$\alpha$ model-agnostic watermarking $(\eta, \{\mathcal{P}_\rho\}_{\rho \in \Delta(\Omega, \mathcal{F})})$, the following holds

$$\sum_{A \in 2^\Omega} \eta(A)\mathbb{1}(x \in A) \leq \alpha, \quad \forall x \in \Omega.$$

For simplicity of notations, we assume $\Omega = \{1, 2, \ldots, m\}$. We consider hard instances in the form of $\text{Unif}(i_1, i_2, \ldots, i_{1/\alpha})$ where $1 \leq i_1 < \cdots < i_{1/\alpha} \leq m$. Notice that for any $\rho_0 =$

$\text{Unif}(i_1, i_2, \ldots, i_{1/\alpha})$, we have $\beta(\mathcal{P}_{\rho_0}^*) = 0$ and

$$\beta(\mathcal{P}_{\rho_0}) \geq \mathbb{P}_{A \sim \eta}\left(\{i_1, \ldots, i_{1/\alpha}\} \cap A = \emptyset\right)$$

$$\geq \sum_A \eta(A) \cdot \prod_{j=1}^{1/\alpha} \mathbb{1}(i_j \notin A).$$

By probabilistic method,

$$\max_{\rho_0} \beta(\mathcal{P}_{\rho_0}) \geq \max_{i_1 < \cdots < i_{1/\alpha}} \sum_A \eta(A) \cdot \prod_{j=1}^{1/\alpha} \mathbb{1}(i_j \notin A)$$

$$\geq \frac{1}{\binom{m}{1/\alpha}} \sum_{i_1 < \cdots < i_{1/\alpha}} \sum_A \eta(A) \cdot \prod_{j=1}^{1/\alpha} \mathbb{1}(i_j \notin A).$$

It follows that the maximum Type II error loss is lower bounded by the optimum $v^*$ of the following linear program

$$v^* = \min_{\eta} \frac{1}{\binom{m}{1/\alpha}} \sum_{i_1 < \cdots < i_{1/\alpha}} \sum_A \eta(A) \cdot \prod_{j=1}^{1/\alpha} \mathbb{1}(i_j \notin A)$$

$$\text{s.t.} \sum_{A \in 2^\Omega} \eta(A) \mathbb{1}(x \in A) \leq \alpha, \ \forall x \in \Omega,$$

$$\sum_{A \in 2^\Omega} \eta(A) = 1, \ \eta(A) \geq 0, \ \forall A \in 2^\Omega.$$

By duality, we have $v^* =$

$$\min_{\eta \geq 0} \max_{\xi, \zeta \geq 0} \frac{1}{\binom{m}{1/\alpha}} \left( \sum_{i_1 < \cdots < i_{1/\alpha}} \sum_A \eta(A) \cdot \prod_{j=1}^{1/\alpha} \mathbb{1}(i_j \notin A) + \sum_x \xi(x) \left( \sum_{A \in 2^\Omega} \eta(A) \mathbb{1}(x \in A) - \alpha \right) \right.$$

$$\left. + \zeta \cdot \left( \sum_{A \in 2^\Omega} \eta(A) - 1 \right) \right)$$

$$= \max_{\xi, \zeta \geq 0} \min_{\eta \geq 0} \frac{1}{\binom{m}{1/\alpha}} \left( \sum_A \eta(A) \cdot \left( \sum_{i_1 < \cdots < i_{1/\alpha}} \prod_{j=1}^{1/\alpha} \mathbb{1}(i_j \notin A) + \sum_x \xi(x) \mathbb{1}(x \in A) + \zeta \right) - \alpha \cdot \sum_x \xi(x) - \zeta \right)$$

$$\geq \min_{\eta \geq 0} \frac{1}{\binom{m}{1/\alpha}} \sum_{l=1}^{m} \sum_{|A|=l} \eta(A) \cdot \left( \binom{m-l}{1/\alpha} + l \cdot \xi^* + \zeta^* \right) - \frac{\alpha m \xi^* + \zeta^*}{\binom{m}{1/\alpha}}$$

where $\xi^* = \binom{m - \alpha m - 1}{1/\alpha - 1}$ and $\zeta^* = 0$.

Define $f(l) := \binom{m-l}{1/\alpha} + l \cdot \xi^* + \zeta^*$. Since the binomial coefficient $\binom{m-l}{1/\alpha}$ is convex and $l \cdot \xi^* + \zeta^*$ is linear, $f$ a convex function. Notice that

$$f(\alpha m) - f(\alpha m - 1) = \xi^* - \binom{m - \alpha m}{1/\alpha - 1}$$

$$< 0$$

and

$$f(\alpha m + 1) - f(\alpha m) = \xi^* - \binom{m - \alpha m - 1}{1/\alpha - 1}$$

$$= 0,$$

$f$ achieves minimum at $l^* = \alpha m$. It follows that

$$\binom{m-l}{1/\alpha} + l \cdot \xi^* + \zeta^* \geq \binom{m - \alpha m}{1/\alpha} + \alpha m \cdot \binom{m - \alpha m - 1}{1/\alpha - 1}$$

holds for all $l \in [m]$ and thus

$$\mathbf{RHS} \geq \frac{\binom{m-\alpha m}{1/\alpha}}{\binom{m}{1/\alpha}} = \frac{\binom{m-1/\alpha}{\alpha m}}{\binom{m}{\alpha m}} = \frac{\binom{m-\frac{1}{\alpha}}{\alpha m}}{\binom{m}{\alpha m}}.$$

**Upper bound.** Define $p$ as the projection from $\Omega \times 2^\Omega$ to $2^\Omega$, i.e. $p(V) = \{A \in 2^\Omega : \exists x \in \Omega, \ s.t.(x, A) \in V\}$. Let $W := \{(x, A) \in \Omega \times 2^\Omega : x \in A\}$.

Notice that the marginal distribution of reject region

$$\eta^*(A) = \begin{cases} \frac{1}{\binom{m}{\alpha m}}, & \text{if } |A| = \alpha m \\ 0, & \text{otherwise} \end{cases}.$$

satisfies

$$\sup_{\pi \in \Delta(\Omega)} \mathbb{P}_{Y \sim \pi, (X,R) \sim \mathcal{P}}(Y \in R) \leq \sum_{x \in \Omega} \pi(x) \cdot \eta^* \left( p(W \cap (\{x\} \times 2^\Omega)) \right)$$

$$\leq \sum_{x \in \Omega} \pi(x) \cdot \left( 1 - \frac{\binom{m-1}{\alpha m}}{\binom{m}{\alpha m}} \right)$$

$$= \alpha.$$

Hence it guarantees Type I error $\leq \alpha$.

It suffices to show (*): for any $\rho \in \Delta(\Omega, \mathcal{F})$, there exists a coupling $\mathcal{P}_\rho$ of $\eta^*$ and $\rho$ such that

$$\mathbb{P}_{(x,A) \sim \mathcal{P}_\rho}(x \notin A) \leq \frac{\binom{m-\frac{1}{\alpha}}{\alpha m}}{\binom{m}{\alpha m}} + \sum_{x:\rho(x) \geq \alpha}(\rho(x) - \alpha).$$

To show the above, we check the Strassen's condition

$$\rho(U) - \eta^* \left( p \left( W \cap (U \times 2^\Omega) \right) \right) \leq \frac{\binom{m-\frac{1}{\alpha}}{\alpha m}}{\binom{m}{\alpha m}} + \sum_{x:\rho(x) \geq \alpha}(\rho(x) - \alpha), \forall U \subset \Omega. \qquad (13)$$

Indeed, given Eq. (13), Theorem 11 in Strassen (1965) establishes (*).

In the rest of the proof, we show Eq. (13). Fix $U$ with cardinality $k$. First notice that

$$\rho(U) - \sum_{x:\rho(x) \geq \alpha}(\rho(x) - \alpha) \leq (\alpha k \wedge 1).$$

Since $p \left( W \cap (U \times 2^\Omega) \right) = \{A \in 2^\Omega : \exists i \in U, \ s.t. \ i \in A\}$, we have

$$\eta^* \left( p \left( W \cap (U \times 2^\Omega) \right) \right) \geq 1 - \frac{\binom{m-k}{\alpha m}}{\binom{m}{\alpha m}} = 1 - \frac{\binom{m-\alpha m}{k}}{\binom{m}{k}}.$$

In the remaining paper, $\binom{m-\alpha m}{k}$ is understood as zero if $m - \alpha m < k$.

If $k \leq \frac{1}{\alpha}$, then because $g(k) := \alpha k - 1 + \frac{\binom{m-\alpha m}{k}}{\binom{m}{k}}$ is convex and takes maximum $\frac{\binom{m-\alpha m}{\frac{1}{\alpha}}}{\binom{m}{\frac{1}{\alpha}}} = \frac{\binom{m-\frac{1}{\alpha}}{\alpha m}}{\binom{m}{\alpha m}}$

at $k^* = \frac{1}{\alpha}$, we have

$$\rho(U) - \eta^* \left( p \left( W \cap (U \times 2^\Omega) \right) \right) \leq \alpha k - 1 + \frac{\binom{m-\alpha m}{k}}{\binom{m}{k}} + \sum_{x:\rho(x) \geq \alpha}(\rho(x) - \alpha)$$

$$= \frac{\binom{m-\frac{1}{\alpha}}{\alpha m}}{\binom{m}{\alpha m}} + \sum_{x:\rho(x) \geq \alpha}(\rho(x) - \alpha).$$

If $k \geq \frac{1}{\alpha}$, then since $\frac{\binom{m-\alpha m}{k}}{\binom{m}{k}} = \frac{\binom{m-k}{\alpha m}}{\binom{m}{\alpha m}}$ is monotonously decreasing in $k$,

$$\rho(U) - \eta^* \left( p \left( W \cap (U \times 2^\Omega) \right) \right) \leq \frac{\binom{m-\alpha m}{k}}{\binom{m}{k}} + \sum_{x:\rho(x)\geq\alpha} (\rho(x) - \alpha)$$

$$= \frac{\binom{m-\frac{1}{\alpha}}{\alpha m}}{\binom{m}{\alpha m}} + \sum_{x:\rho(x)\geq\alpha} (\rho(x) - \alpha).$$

Combining, we establishes Eq. (13).

Under the condition $\alpha \to 0_+$ and $1/(\alpha m) \to 0_+$, the rate displayed in Theorem 1 simplifies to:

$$\frac{(m - \alpha m)(m - \alpha m - 1) \cdots (m - \alpha m - 1/\alpha + 1)}{n(m-1) \cdots (m - 1/\alpha + 1)} \asymp (1 - \alpha)^{1/\alpha} \to e^{-1}.$$

This concludes the proof. $\qquad\square$

### G.4 Proof of Theorem C.8

*Proof.* We follow the notations in the proof of Theorem C.6.

**Lower bound.** By definition of Type I error, for any level-$\alpha$ model-agnostic watermarking $(\eta, \{\mathcal{P}_\rho\}_{\rho\in\Delta(\Omega,\mathcal{F})})$, the following holds

$$\sum_{A\in 2^\Omega} \eta(A)\mathbb{1}(x \in A) \leq \alpha, \ \forall x \in \Omega.$$

We consider hard instances in the form of $\mathrm{Unif}(i_1, i_2, \ldots, i_{1/\kappa})$ where $1 \leq i_1 < \cdots < i_{1/\kappa} \leq m$. Notice that for any $\rho_0 = \mathrm{Unif}(i_1, i_2, \ldots, i_{1/\kappa})$, we have $\beta(\mathcal{P}^*_{\rho_0}) = 0$ and

$$\beta(\mathcal{P}_{\rho_0}) \geq \mathbb{P}_{A\sim\eta} \left( \{i_1, \ldots, i_{1/\kappa}\} \cap A = \emptyset \right)$$

$$\geq \sum_A \eta(A) \cdot \prod_{j=1}^{1/\kappa} \mathbb{1}(i_j \notin A).$$

By probabilistic method,

$$\max_{\rho_0} \beta(\mathcal{P}_{\rho_0}) \geq \max_{i_1<\cdots<i_{1/\kappa}} \sum_A \eta(A) \cdot \prod_{j=1}^{1/\kappa} \mathbb{1}(i_j \notin A)$$

$$\geq \frac{1}{\binom{m}{1/\kappa}} \sum_{i_1<\cdots<i_{1/\kappa}} \sum_A \eta(A) \cdot \prod_{j=1}^{1/\kappa} \mathbb{1}(i_j \notin A).$$

It follows that the maximum Type II error loss is lower bounded by the optimum $v^*$ of the following linear program

$$v^* = \min_\eta \frac{1}{\binom{m}{1/\kappa}} \sum_{i_1<\cdots<i_{1/\kappa}} \sum_A \eta(A) \cdot \prod_{j=1}^{1/\kappa} \mathbb{1}(i_j \notin A)$$

$$\text{s.t.} \sum_{A\in 2^\Omega} \eta(A)\mathbb{1}(x \in A) \leq \alpha, \ \forall x \in \Omega,$$

$$\sum_{A\in 2^\Omega} \eta(A) = 1, \ \eta(A) \geq 0, \ \forall A \in 2^\Omega.$$

By duality, we have $v^* =$

$$\min_{\eta \geq 0} \max_{\xi, \zeta \geq 0} \frac{1}{\binom{m}{1/\kappa}} \left( \sum_{i_1 < \cdots < i_{1/\kappa}} \sum_A \eta(A) \cdot \prod_{j=1}^{1/\kappa} \mathbb{1}(i_j \notin A) + \sum_x \xi(x) \left( \sum_{A \in 2^\Omega} \eta(A) \mathbb{1}(x \in A) - \alpha \right) \right.$$

$$\left. + \zeta \cdot \left( \sum_{A \in 2^\Omega} \eta(A) - 1 \right) \right)$$

$$= \max_{\xi, \zeta \geq 0} \min_{\eta \geq 0} \frac{1}{\binom{m}{1/\kappa}} \left( \sum_A \eta(A) \cdot \left( \sum_{i_1 < \cdots < i_{1/\kappa}} \prod_{j=1}^{1/\kappa} \mathbb{1}(i_j \notin A) + \sum_x \xi(x) \mathbb{1}(x \in A) + \zeta \right) - \alpha \cdot \sum_x \xi(x) - \zeta \right)$$

$$\geq \min_{\eta \geq 0} \frac{1}{\binom{m}{1/\kappa}} \sum_{l=1}^m \sum_{|A|=l} \eta(A) \cdot \left( \binom{m-l}{1/\kappa} + l \cdot \xi^* + \zeta^* \right) - \frac{\alpha m \xi^* + \zeta^*}{\binom{m}{1/\kappa}}$$

where $\xi^* = \binom{m-\alpha m-1}{1/\kappa-1}$ and $\zeta^* = 0$.

Define $f(l) := \binom{m-l}{1/\kappa} + l \cdot \xi^* + \zeta^*$. Since the binomial coefficient $\binom{m-l}{1/\kappa}$ is convex and $l \cdot \xi^* + \zeta^*$ is linear, $f$ a convex function. Notice that

$$f(\alpha m) - f(\alpha m - 1) = \xi^* - \binom{m - \alpha m}{1/\kappa - 1}$$

$$< 0$$

and

$$f(\alpha m + 1) - f(\alpha m) = \xi^* - \binom{m - \alpha m - 1}{1/\kappa - 1}$$

$$= 0,$$

$f$ achieves minimum at $l^* = \alpha m$. It follows that

$$\binom{m-l}{1/\kappa} + l \cdot \xi^* + \zeta^* \geq \binom{m - \alpha m}{1/\kappa} + \alpha m \cdot \binom{m - \alpha m - 1}{1/\kappa - 1}$$

holds for all $l \in [m]$ and thus

$$\textbf{RHS} \geq \frac{\binom{m-\alpha m}{1/\kappa}}{\binom{m}{1/\kappa}} = \frac{\binom{m-1/\kappa}{\alpha m}}{\binom{m}{\alpha m}} = \frac{\binom{m-\frac{1}{\kappa}}{\alpha m}}{\binom{m}{\alpha m}}.$$

**Upper bound.** Notice that the marginal distribution of reject region

$$\eta^*(A) = \begin{cases} \frac{1}{\binom{m}{\alpha m}}, & \text{if } |A| = \alpha m \\ 0, & \text{otherwise} \end{cases}.$$

satisfies

$$\sup_{\pi \in \Delta(\Omega)} \mathbb{P}_{Y \sim \pi, (X,R) \sim \mathcal{P}}(Y \in R) \leq \sum_{x \in \Omega} \pi(x) \cdot \eta^* \left( p(W \cap (\{x\} \times 2^\Omega)) \right)$$

$$\leq \sum_{x \in \Omega} \pi(x) \cdot \left( 1 - \frac{\binom{m-1}{\alpha m}}{\binom{m}{\alpha m}} \right)$$

$$= \alpha.$$

Hence it guarantees Type I error $\leq \alpha$.

In what remains, we define $p$ and $W$ in the same way in the proof of Theorem C.6 and check the Strassen's condition

$$\rho(U) - \eta^* \left( p \left( W \cap (U \times 2^\Omega) \right) \right) \leq \frac{\binom{m-\frac{1}{\alpha}}{\alpha m}}{\binom{m}{\alpha m}} + \sum_{x: \rho(x) \geq \alpha} (\rho(x) - \alpha), \forall U \subset \Omega. \tag{14}$$

Fix $U$ with cardinality $k$. Due to the condition of $\sup_{\omega \in \Omega} \rho(\{\omega\}) \leq \kappa$, we have $\rho(U) - \sum_{x: \rho(x) \geq \alpha} (\rho(x) - \alpha) \leq (\kappa k \wedge 1)$. Since $p \left( W \cap (U \times 2^\Omega) \right) = \{A \in 2^\Omega : \exists i \in U, \ s.t. \ i \in A\}$, we

have

$$\eta^* \left( p \left( W \cap (U \times 2^\Omega) \right) \right) \geq 1 - \frac{\binom{m-k}{\alpha m}}{\binom{m}{\alpha m}} = 1 - \frac{\binom{m-\alpha m}{k}}{\binom{m}{k}}.$$

If $k \leq \frac{1}{\kappa}$, then

$$\rho(U) - \eta^* \left( p \left( W \cap (U \times 2^\Omega) \right) \right) \leq \kappa k - 1 + \frac{\binom{m-\alpha m}{k}}{\binom{m}{k}} + \sum_{x:\rho(x)\geq\alpha} (\rho(x) - \alpha)$$

$$= \frac{\binom{m-\frac{1}{\kappa}}{\alpha m}}{\binom{m}{\alpha m}} + \sum_{x:\rho(x)\geq\alpha} (\rho(x) - \alpha).$$

where the second step follows from the fact that $g(k) := \kappa k - 1 + \frac{\binom{m-\alpha m}{k}}{\binom{m}{k}}$ is convex and takes

maximum $\frac{\binom{m-\alpha m}{\frac{1}{\kappa}}}{\binom{m}{\frac{1}{\kappa}}} = \frac{\binom{m-\frac{1}{\kappa}}{\alpha m}}{\binom{m}{\alpha m}}$ at $k^* = \frac{1}{\kappa}$.

If $k \geq \frac{1}{\kappa}$, then

$$\rho(U) - \eta^* \left( p \left( W \cap (U \times 2^\Omega) \right) \right) \leq \frac{\binom{m-\alpha m}{k}}{\binom{m}{k}} + \sum_{x:\rho(x)\geq\alpha} (\rho(x) - \alpha)$$

$$= \frac{\binom{m-\frac{1}{\kappa}}{\alpha m}}{\binom{m}{\alpha m}} + \sum_{x:\rho(x)\geq\alpha} (\rho(x) - \alpha)$$

where the inequality is because $\frac{\binom{m-\alpha m}{k}}{\binom{m}{k}} = \frac{\binom{m-k}{\alpha m}}{\binom{m}{\alpha m}}$ is monotonously decreasing in $k$. Combining, we

establishes Eq. (14).

Combining the above cases, we checked Strassen's condition and hence the statement follows. $\square$

### G.5 PROOF OF THEOREM C.14

*Proof.* Let $\kappa_0 = \alpha / \log \frac{1}{\beta}$, $\kappa := e^{-\widehat{H}} \lesssim \kappa_0$, and $m := |\Omega| = |\Omega_0|^n \gtrsim 1/\kappa_0$. Define $p$ as the projection from $\Omega \times 2^\Omega$ to $2^\Omega$: $\forall V$, $p(V) = \{A \in 2^\Omega : \exists x \in \Omega,\ s.t.\ (x, A) \in V\}$. Define $\bar{\Omega} := \{x \in \Omega : \rho(x) \leq \kappa\}$, $W := \{(x, A) \in \Omega \times 2^\Omega : x \in A\}$, and $\bar{W} := \{(x, A) \in \bar{\Omega} \times 2^\Omega : x \in A\}$. Notice that

$$\sup_{\pi \in \Delta(\Omega)} \mathbb{P}_{Y\sim\pi, (X,R)\sim\mathcal{P}}(Y \in R) \leq \sum_{x\in\Omega} \pi(x) \cdot \eta^* \left( p(W \cap (\{x\} \times 2^\Omega)) \right)$$

$$\leq \sum_{x\in\Omega} \pi(x) \cdot \left( 1 - \frac{\binom{m-1}{\alpha m}}{\binom{m}{\alpha m}} \right) = \alpha$$

thus Type I error $\leq \alpha$. To establish the conditional Type II error guarantee, we check the following Strassen's condition, similarly to Theorem 3.6,

$$\rho(U) - \eta^* \left( p \left( \bar{W} \cap (U \times 2^\Omega) \right) \right) \leq \rho(\bar{\Omega}^c) + \beta \cdot \rho(\bar{\Omega}), \forall U \subset \Omega. \tag{15}$$

Fix $U$ and define $k := |U \cap \bar{\Omega}|$. Notice that $\rho(U) \leq (\kappa k + \rho(\bar{\Omega}^c)) \wedge 1$. It follows that

$$\rho(U) - \eta^* \left( p \left( \bar{W} \cap (U \times 2^\Omega) \right) \right) \leq \left( (\kappa k + \rho(\bar{\Omega}^c)) \wedge 1 \right) - 1 + \frac{\binom{m-k}{\alpha m}}{\binom{m}{\alpha m}}$$

$$\leq \max \left\{ \frac{\binom{m-\frac{\rho(\bar{\Omega})}{\kappa}}{\alpha m}}{\binom{m}{\alpha m}} - \rho(\bar{\Omega}^c), 0 \right\} + \rho(\bar{\Omega}^c).$$

where the second step follows from the fact that $\left( (\kappa k + \rho(\bar{\Omega}^c)) \wedge 1 \right) - 1 + \frac{\binom{m-k}{\alpha m}}{\binom{m}{\alpha m}}$ is convex in

$[0, \rho(\bar{\Omega})/\kappa]$ and decreasing in $[\rho(\bar{\Omega})/\kappa, \infty]$, and thus the maximum can only be taken at either $k = 0$ or $k = \rho(\bar{\Omega})/\kappa$.

By the conditions of $n$ and $\kappa$, some arithmetic shows that

$$\max\left\{\frac{\binom{m-\frac{\rho(\bar{\Omega})}{\kappa}}{\alpha m}}{\binom{m}{\alpha m}}-\rho(\bar{\Omega}^c),0\right\} \leq \max\left\{\frac{\binom{m-\frac{\rho(\bar{\Omega})}{\kappa_0}}{\alpha m}}{\binom{m}{\alpha m}}-\rho(\bar{\Omega}^c),0\right\}$$

$$\lesssim \max\left\{(1-\alpha)^{\rho(\bar{\Omega})/\kappa_0}-1+\rho(\bar{\Omega}),0\right\}$$

$$\lesssim \max\left\{\beta^{\rho(\bar{\Omega})}-1+\rho(\bar{\Omega}),0\right\}$$

$$\leq \beta \cdot \rho(\bar{\Omega})$$

where the first inequality comes from $\kappa \leq \kappa_0$, the second inequality is because

$$\frac{\binom{m-\frac{\rho(\bar{\Omega})}{\kappa_0}}{\alpha m}}{\binom{m}{\alpha m}} = \frac{(m-\alpha m)(m-\alpha m-1)\cdots(m-\alpha m-\frac{\rho(\bar{\Omega})}{\kappa_0}+1)}{m(m-1)\cdots(m-\frac{\rho(\bar{\Omega})}{\kappa_0}+1)}$$

$$\lesssim (1-\alpha)^{\rho(\bar{\Omega})/\kappa_0}$$

due to $m \gtrsim 1/\kappa_0$, the third inequality follows from $\kappa_0 = \alpha/\log(1/\beta)$, and the last inequality is due to the observation that $\max\left\{\beta^{\rho(\bar{\Omega})}-1+\rho(\bar{\Omega}),0\right\}-\beta\cdot\rho(\bar{\Omega})$ is a convex function of $\rho(\bar{\Omega})$ and takes maximum at either $\rho(\bar{\Omega})=0$ or $1$.

Eq. (15) hence follows. Applying Strassen's Theorem (Strassen, 1965), we have $\mathbb{P}_{(X,R)\sim\mathcal{P}}(\bar{W}) \geq 1-\left(\rho(\bar{\Omega}^c)+\beta\cdot\rho(\bar{\Omega})\right) = \rho(\bar{\Omega})\cdot(1-\beta)$.

By Bayes' law, $\mathbb{P}_{(X,R)\sim\mathcal{P}}(X \notin R|-\log\rho(X) \geq \widehat{H}) = \mathbb{P}_{(X,R)\sim\mathcal{P}}(X \notin R|\rho(X) \leq \kappa) = \frac{1-\mathbb{P}_{(X,R)\sim\mathcal{P}}(\bar{W})}{\rho(\bar{\Omega})} \leq \beta$. This completes the proof. $\square$

### G.6 Proof of Theorem C.18

*Proof.* Throughout the proof we omit the subscript in the shrinkage operator $\mathcal{S}$, as $G$ is fixed. First notice that

$$\mathbb{E}_{X,R\sim\mathcal{P}}\left[\min_{Y\in out(X)}\mathbb{1}(Y \in R)\right] = \mathcal{P}(X \in \mathcal{S}(R))$$

$$= \sum_{y\in\Omega}\sum_{R\in 2^\Omega}\rho(y)\mathcal{P}(R|y)\mathbb{1}(y \in \mathcal{S}(R)).$$

Further, notice that $y \in in(z)$ and $y \in \mathcal{S}(R)$ implies that $z \in R$, thus

$$\sum_{y\in in(z)}\sum_{R\in 2^\Omega}\rho(y)\mathcal{P}(R|y)\mathbb{1}(y \in \mathcal{S}(R)) \leq \sum_{y\in in(z)}\sum_{R\in 2^\Omega}\rho(y)\mathcal{P}(R|y)\mathbb{1}(z \in R)$$

$$\leq \sum_{y\in\Omega}\sum_{R\in 2^\Omega}\rho(y)\mathcal{P}(R|y)\mathbb{1}(z \in R)$$

$$= \mathbb{P}_{X\sim\delta_z,R\sim\mathcal{P}(\Omega,\cdot)}(X \in R)$$

$$\leq \alpha.$$

It follows that the optimum Type II error is lower bounded by the optimum of the following Linear Program

$$\min_{\mathcal{P}} 1 - \sum_{y\in\Omega}\sum_{R\in 2^\Omega}\rho(y)\mathcal{P}(R|y)\mathbb{1}(y \in \mathcal{S}(R)) \tag{16}$$

$$s.t. \sum_{y\in in(z)}\sum_{R\in 2^\Omega}\rho(y)\mathcal{P}(R|y)\mathbb{1}(y \in \mathcal{S}(R)) \leq \alpha, \sum_{R\in 2^\Omega}\mathcal{P}(R|z)=1, 0\leq\mathcal{P}(R|z)\leq 1, \forall z\in\Omega, R\in 2^\Omega.$$

We claim that the minimum in Eq. (16) is equal to the minimum of Eq. (2). Indeed, it suffices to show that Eq. (16) is optimized when $\mathcal{P}(\cdot|y_0)$ is supported on $\{\emptyset, \mathcal{S}^{-1}(\{y_0\})\}$ (then setting $x(y) \equiv \mathcal{P}(\mathcal{S}^{-1}(\{y\})|y)$ reduces Eq. (16) to Eq. (2)). To see this, consider any minimizer $\widetilde{\mathcal{P}}$ such that there exists $y_0 \in \Omega$ and $R_0 \notin \{\emptyset, \mathcal{S}^{-1}(\{y_0\})\}$, with $\widetilde{\mathcal{P}}(R_0|y_0) > 0$. We will show that there exists $\bar{\mathcal{P}}$ such

that it achieves the no greater objective value, and satisfies $|\text{supp}(\bar{\mathcal{P}}(\cdot|y_0)) \cap \{\emptyset, \mathcal{S}^{-1}(\{y_0\})\}^c| = |\text{supp}(\widetilde{\mathcal{P}}(\cdot|y_0)) \cap \{\emptyset, \mathcal{S}^{-1}(\{y_0\})\}^c| - 1$ and $|\text{supp}(\bar{\mathcal{P}}(\cdot|y))| = |\text{supp}(\widetilde{\mathcal{P}}(\cdot|y))|$ for all other $y \in \Omega$. Iteratively applying this argument, we reduce $\text{supp}(\widetilde{\mathcal{P}}(\cdot|y)) \cap \{\emptyset, \mathcal{S}^{-1}(\{y\})\}^c$ to $\emptyset$ for any $y \in \Omega$ and thereby prove the claim.

Consider the following two cases.

**Case 1:** $y_0 \notin \mathcal{S}(R_0)$. Then letting

$$\bar{\mathcal{P}}(R|y) = \begin{cases} \widetilde{\mathcal{P}}(R_0|y) + \widetilde{\mathcal{P}}(R|y), & y = y_0, R = \emptyset \\ 0, & y = y_0, R = R_0 \\ \widetilde{\mathcal{P}}(R|y), & \text{o.w.,} \end{cases}$$

we observe that

$$\sum_{y \in \Omega} \sum_{R \in 2^\Omega} \rho(y)\widetilde{\mathcal{P}}(R|y)\mathbb{1}(y \in \mathcal{S}(R)) = \sum_{y \in \Omega} \sum_{R \in 2^\Omega} \rho(y)\bar{\mathcal{P}}(R|y)\mathbb{1}(y \in \mathcal{S}(R))$$

and $\bar{\mathcal{P}}$ satisfies all the constraints in Eq. (16). It is obvious from the construction of $\bar{\mathcal{P}}$ that $|\text{supp}(\bar{\mathcal{P}}(\cdot|y_0)) \cap \{\emptyset, \mathcal{S}^{-1}(\{y_0\})\}^c| = |\text{supp}(\widetilde{\mathcal{P}}(\cdot|y_0)) \cap \{\emptyset, \mathcal{S}^{-1}(\{y_0\})\}^c| - 1$ and $|\text{supp}(\bar{\mathcal{P}}(\cdot|y))| = |\text{supp}(\widetilde{\mathcal{P}}(\cdot|y))|$ for all other $y \in \Omega$.

**Case 2:** $y_0 \in \mathcal{S}(R_0)$. Then letting

$$\bar{\mathcal{P}}(R|y) = \begin{cases} \widetilde{\mathcal{P}}(R_0|y) + \widetilde{\mathcal{P}}(R|y), & y = y_0, R = \mathcal{S}^{-1}(\{y_0\}) \\ 0, & y = y_0, R = R_0 \\ \widetilde{\mathcal{P}}(R|y), & \text{o.w.} \end{cases},$$

we observe that

$$\sum_{y \in \Omega} \sum_{R \in 2^\Omega} \rho(y)\widetilde{\mathcal{P}}(R|y)\mathbb{1}(y \in \mathcal{S}(R)) = \sum_{y \in \Omega} \sum_{R \in 2^\Omega} \rho(y)\bar{\mathcal{P}}(R|y)\mathbb{1}(y \in \mathcal{S}(R))$$

and $\bar{\mathcal{P}}$ satisfies all the constraints in Eq. (16) due to $\mathbb{1}(y \in \mathcal{S}(R_0)) \geq \mathbb{1}(y \in \mathcal{S}(\{y_0\}))$ for any $y \in \Omega$. From the construction of $\bar{\mathcal{P}}$, we know that $|\text{supp}(\bar{\mathcal{P}}(\cdot|y_0)) \cap \{\emptyset, \mathcal{S}^{-1}(\{y_0\})\}^c| = |\text{supp}(\widetilde{\mathcal{P}}(\cdot|y_0)) \cap \{\emptyset, \mathcal{S}^{-1}(\{y_0\})\}^c| - 1$ and $|\text{supp}(\bar{\mathcal{P}}(\cdot|y))| = |\text{supp}(\widetilde{\mathcal{P}}(\cdot|y))|$ for all other $y \in \Omega$.

Combining the above cases, we established our claim.

Finally, letting $\mathcal{P}^*(\cdot|y) = x^*(y) \cdot \delta_{\mathcal{S}^{-1}(\{y\})}$ for all $y \in \omega$, where $x^*$ is the solution of Eq. (2), achieves the optimum value in Eq. (2). $\qquad\square$

### G.7 PROOF OF THEOREM 3.1 AND THEOREM 3.2

Throughout this section, we will use $\rho_j(\cdot)$ and $q_j(\cdot)$ to abbreviate $\rho_j(\cdot|p, t_{1:j-1})$ and $q_j(\cdot|t_{j-K:j-1})$ respectively. For statistical analysis, we will also assume the pseudo-randomness functions used in Algorithm 3-4 are true random oracles. Our statistical results can be transformed into cryptography results by hardness hypothesis on the pseudo-random functions.

**Theorem G.4** (Distortion-free). *The watermark in Algorithm 3 is distortion-free. More precisely, for any sequence $t_{1:j-1}$ and any token $w$,*

$$\mathbb{P}_{\text{SEAL}}(t_j = w|p, t_{1:j-1}) = \rho_j(w|p, t_{1:j-1}).$$

*where $\mathbb{P}_{\text{SEAL}}$ denotes the next-token probability under the SEAL generation phase, $p$ is the prompt, and $\rho_j$ represents the original model's next-token distribution.*

*Proof.* Fix $h_j$. Let $\mu_{S;j}$ and $\mu_j$ denote $h_j \# q_j$ and $h_j \# \rho_j$ respectively. Since Algorithm 3 performs a maximal coupling between $\mu_{S;j}$ and $\mu_j$, it follows directly that (see e.g. Leviathan et al. (2023) for detailed derivation)

$$\mathbb{P}_{\text{SEAL}}(s_j = w|p, t_{1:j-1}, h_j) = \mu_j(w|p, t_{1:j-1}).$$

Therefore

$$\mathbb{P}_{\text{SEAL}}(t_j = u|p, t_{1:j-1}, h_j) = \sum_w \mathbb{P}_{\text{SEAL}}(s_j = w|p, t_{1:j-1}, h_j) \cdot \mathbb{P}_{\text{SEAL}}(t_j = u|h_j(t_j) = s_j, h_j)$$

$$= \sum_w \mu_j(w|p, t_{1:j-1}) \cdot \rho_j(t_j = u|h_j(t_j) = w)$$

$$= \sum_w h_j \# \rho_j(w|p, t_{1:j-1}) \cdot \rho_j(t_j = u|h_j(t_j) = w)$$

$$= \rho_j(u|p, t_{1:j-1}).$$

Taking total probability for independent random $h_j$, we have

$$\mathbb{P}_{\text{SEAL}}(t_j = w|p, t_{1:j-1}) = \sum_{h_j} \mathbb{P}_{\text{SEAL}}(h_j|p, t_{1:j-1}) \cdot \mathbb{P}_{\text{SEAL}}(t_j = w|p, t_{1:j-1}, h_j)$$

$$= \sum_h \mathbb{P}_{\text{SEAL}}(h_j|p, t_{1:j-1}) \cdot \rho_j(w|p, t_{1:j-1})$$

$$= \rho_j(w|p, t_{1:j-1}).$$

$\square$

**Theorem G.5** (False positive control). *For any fixed (sub-)sequence $y$,*

$$\mathbb{P}_{\text{SEAL}}(\texttt{watermarked} = \text{True } \textit{for sequence } y) \le \alpha.$$

*where $\mathbb{P}_{\text{SEAL}}$ denotes the randomness in SEAL detection phase.*

*Proof.* For any fixed (sub-)sequence $y$, $\xi_j$'s are i.i.d. Bernoulli random variables with

$$\mathbb{P}(\xi_j = 1) = \mathbb{P}(h_j(s_j) = h_j(t_j)) = h_j \sharp q_j(h_j(t_j)).$$

If the threshold is computed using Eq. (4), then since

$$\mathbb{P}(Z_i = 1) = w_i, \ \mathbb{P}(Z_i = 0) = 1 - w_i,$$

where $w_j = h_j \sharp q_j(h_j(t_j)) = \mathbb{P}(Z_j = 1)$, we have

$$\mathbb{P}\left(\sum_{j=i}^{i+k} Z_j = l\right) = \mathbb{P}(Z_{i+k} = 1) \cdot \mathbb{P}\left(\sum_{j=i}^{i+k-1} Z_j = l - 1\right) + \mathbb{P}(Z_{i+k} = 0) \cdot \mathbb{P}\left(\sum_{j=i}^{i+k-1} Z_j = l\right)$$

$$= w_{i+k} \cdot \mathbb{P}\left(\sum_{j=i}^{i+k-1} Z_j = l - 1\right) + (1 - w_{i+k}) \cdot \mathbb{P}\left(\sum_{j=i}^{i+k-1} Z_j = l\right)$$

It follows by induction that the $p_{k,l}$'s computed by Eq. (4) satisfy $p_{k,l} = \mathbb{P}\left(\sum_{j=i}^{i+k-1} Z_j = l\right)$.
Therefore $\mathbb{P}_{\text{SEAL}}(\texttt{watermarked} = \text{True}) \le \alpha$ holds by definition.

Next, we consider the case of using Eq. (5). Define

$$\mu = \sum_{j=i}^{i+L} h \# q_j(h_j(t_j)).$$

By Lemma G.6 and choice of $\epsilon$, we have

$$\mathbb{P}\left(\sum_{j=i}^{i+L} \xi_j \ge (1+\epsilon)\mu\right) \le e^{(\epsilon - (1+\epsilon)\log(1+\epsilon))\mu}$$

$$\le \alpha.$$

It follows that

$$\mathbb{P}_{\text{SEAL}}(\texttt{watermarked} = \text{True}) = \mathbb{P}\left(\sum_{j=i}^{i+L} \xi_j \ge (1+\epsilon)\mu\right)$$

$$\le \alpha.$$

$\square$

**Lemma G.6.** *Let $X$ be the sum of independent Bernoulli random variables (not necessarily with the same mean). Let $\mu = \mathbb{E}[X]$. Then for all $\epsilon > 0$,*

$$\mathbb{P}(X \geq (1+\epsilon)\mu) \leq e^{(\epsilon - (1+\epsilon)\log(1+\epsilon))\mu}.$$

*Proof.* Letting $X = \sum_{i=1}^{n} X_i$ where $X_i \sim \text{Bernoulli}(p_i)$. By Chernoff bound,

$$\mathbb{P}(X \geq (1+\epsilon)\mu) \leq \frac{\mathbb{E}\left[e^{\lambda X}\right]}{e^{(1+\epsilon)\mu\lambda}}$$

$$= \frac{\prod_{i=1}^{n} \mathbb{E}\left[e^{\lambda X_i}\right]}{e^{(1+\epsilon)\mu\lambda}}$$

By MGF of Bernoulli distribution,

$$\frac{\prod_{i=1}^{n} \mathbb{E}\left[e^{\lambda X_i}\right]}{e^{(1+\epsilon)\mu\lambda}} = e^{\sum_{i=1}^{n} \log(p_i e^\lambda + 1 - p_i) - (1+\epsilon)\mu\lambda}$$

$$\leq e^{\sum_{i=1}^{n}(p_i e^\lambda - p_i) - (1+\epsilon)\mu\lambda}$$

$$= e^{\left(e^\lambda - 1 - (1+\epsilon)\lambda\right)\mu}$$

where the inequality applies the hint.

Since $e^\lambda - 1 - (1+\epsilon)\lambda \leq \epsilon - (1+\epsilon)\log(1+\epsilon)$ where the maximum is achieved at $\lambda^* = \log(1+\epsilon)$, we have

$$\mathbb{P}(X \geq (1+\epsilon)\mu) \leq e^{(\epsilon - (1+\epsilon)\log(1+\epsilon))\mu}.$$

$\square$

