# OpenReview forum: "Watermarking using Semantic-aware Speculative Sampling: from Theory to Practice"
_ICLR.cc/2025/Conference — Submitted to ICLR 2025_

### Official Review · Reviewer_xR9M · 2024-10-27

**Soundness:** 3
**Presentation:** 2
**Contribution:** 2
**Rating:** 5
**Confidence:** 4

**Summary:**

This paper investigates methods for embedding watermarks in the outputs of large language models (LLMs). The study examines two types of errors in watermark detection: the first type being false positive errors, and the second type being false negative errors. The paper theoretically discusses the lower bound of the second type of error under the condition of a fixed first type of error. Furthermore, it considers the amount of output text required for embedding the watermark. Finally, the paper presents a construction for embedding watermarks in LLM-generated texts. This construction extracts the semantic information of the output text and uses this semantic information as the seed for generating the watermark, employing speculative sampling to embed the watermark into the subsequent output text.

**Strengths:**

1.The paper attempts to discuss the relationship between two conflicting errors in watermark detection, establishing a lower bound for the second type of error when the first type of error is fixed. This theoretical research is helpful for designing more effective watermark algorithms.

2.The paper attempts to use semantic information as the seed for the watermark, which could be an interesting idea. However, I am not sure what advantages this approach has over directly using the text as the seed.

**Weaknesses:**

1.The paper's construction uses semantic information generated by an LLM as the seed for generating the watermark. However, LLMs are generally probabilistic algorithms, meaning that even with the same prompt input, the outputs can vary between two different instances. If different texts are used as seeds, the resulting pseudo-random numbers will differ, leading to different secret keys as in Fig.2 and making it impossible to complete the watermark detection.

2.The paper's discussion does not consider the impact of an attacker modifying the text. Even slight changes to the text by an attacker could affect the extraction of semantic information, which in turn would impact the watermark detection.

3.The writing in the paper is not clear enough, as the embedding and extraction of the watermark are not explicitly expressed in algorithmic form, making it difficult to understand the construction details.

**Questions:**

1. If the extracted semantic information is not completely consistent, will it result in watermark inconsistency? Will it affect the watermark detection?

2. On page 6,  h_i is a hash function, and in the subsequent description, there is an expression  h_i^{-1}(A) . What does  h_i^{-1}  mean? This form is generally considered an inverse function, but hash functions typically do not have inverse functions.

---

> ### Author Response · Authors · 2024-11-26
> **Response by Authors**
>
> Thank you for your feedback. Please find our detailed responses below.
> > If different texts are used as seeds, the resulting pseudo-random numbers will differ, leading to different secret keys as in Fig.2 and making it impossible to complete the watermark detection.
>
> This is a thoughtful argument: you point out the fundamental difficulty of statistical watermarking, that the (pseudo-)randomness source can be distorted by attack. We would like to clarify that **our contribution to statistical watermarking is orthogonal to the challenge in (pseudo-)randomness source as you pointed out**. Specifically, our work focuses on improving the statistical aspect, i.e. the distribution of the random seeds and the coupling between the random seed and the next token.
>
> Previous works address this challenge by setting the secret keys as keyed hash of the sliding window (usually size <= 6) of past tokens or minimum of keyed hash applied to each previous token in the sliding window. In our experiments, we set the sliding window size to 1, and so the secret keys keep the same given that the latest tokens are not changed. Empirically, we find that **despite paraphrasing, SEAL can still detect nearly half of the watermarked texts (q.v. Table 2)**. Therefore, it is still possible to complete the watermark detection even if different texts are used as seeds.
>
> > The paper's discussion does not consider the impact of an attacker modifying the text. Even slight changes to the text by an attacker could affect the extraction of semantic information, which in turn would impact the watermark detection.
>
> As we clarified above, we experimentally demonstrated that SEAL can still detect nearly half of the watermarked texts under paraphrasing attack.
>
> In our theoretical treatments, we also studied the statistical limits of watermarking under attacks. Specifically, **Appendix C.5 shows that the optimal robust Type II error is governed by a linear program whose constraints correspond to the attacks**.
>
> It is reasonable to expect that a statistical watermark would fail if an attacker significantly modifies the text to the extent that it becomes more reflective of their own work than that of the LLM. If such modifications alter the semantic meaning of the text, the watermark detection is naturally expected to be impacted.
>
> In summary, **the ability of an attacker to invalidate watermark detection by substantially altering the text is not a weakness of our approach but an inherent property of any watermarking system**. Our paper addresses these challenges both theoretically and experimentally, making this critique not a flaw of our work.
>
> > The writing in the paper is not clear enough, as the embedding and extraction of the watermark are not explicitly expressed in algorithmic form, making it difficult to understand the construction details.
>
> Due to space constraints, algorithm pseudocodes are provided in Appendix D.
>
> The embedding and extraction of watermark are discussed in Section 3.1.2 and 3.1.3, simply put, we embed watermark via speculative sampling on a proposal model and extract watermark by checking if the true tokens collide with the tokens from the proposal model.
>
> > If the extracted semantic information is not completely consistent, will it result in watermark inconsistency? Will it affect the watermark detection?
>
> If an attacker changes the semantic information, then our scheme might not detect the watermark. However, as we discussed above, it is fair to expect any watermark to fail if an attacker significantly modifies the text to the extent that it becomes more reflective of their own work than that of the LLM.
>
> > On page 6, h_i is a hash function, and in the subsequent description, there is an expression h_i^{-1}(A) . What does h_i^{-1} mean?
> This form is generally considered an inverse function, but hash functions typically do not have inverse functions.
>
> The preimage of a function is the set of all elements of the domain that are mapped into a given subset of the codomain, which is well-defined for hash functions that are not injective. More precisely, $h_i^{-1}(A) = \{x: h(x) \in A\}$. The preimage (which maps a subset to a subset) can be viewed as a “generalization” of the inverse function (which maps an element to an element).

---

### Official Review · Reviewer_dDvN · 2024-10-29

**Soundness:** 3
**Presentation:** 3
**Contribution:** 3
**Rating:** 6
**Confidence:** 3

**Summary:**

This work makes two contributions: (i) a rigorous theoretical analysis of the statistical limits of watermarking and (ii) a new watermarking scheme named SEAL based on a maximal coupling between the random seed and NTP. SEAL was shown to be more efficient and tamper-resistant than some existing methods in the literature.

**Strengths:**

1. This work provides the first rigorous analysis of the fundamental limit of statistical watermarking. It addresses a few important questions of watermarking that are of theoretical interest.
2. The use of speculative sampling appears to be new and interesting in the current context.
3. The SEAL algorithms have been demonstrated to perform well compared to SOTA methods.

**Weaknesses:**

The paper can be strengthened by presenting the theoretical results in more realistic settings and elaborating more on how these results can be practically relevant. For instance, according to Theorem C.2, the UMP requires the rejection region $R$ to be a singleton set. Such a scheme is practically less useful (it is essentially a retrieval-based approach that involves keeping a growing record of all generated texts and checking for matches). The i.i.d assumption in Theorem 2.3 again makes the results impractical, as it does not hold in any real-world setting. Furthermore, the theoretical results ignore the prompts' impacts, which can significantly affect the next token distribution.
The paper lacks a detailed discussion of the proposed SEAL method's rationale and novelty. It would be particularly helpful to discuss its rationale and compare it with existing semantic-aware watermark schemes in the literature.

**Questions:**

1. Can the authors elaborate more about the role of the proposal LLM in their framework and how its quality affects SEAL's performance?
2. When the proposal LLM is the target LLM and $\Omega_h=${0,1}, it appears that SEAL would reduce to the "green-red" list scheme in Kirchenbauer et al. (2023) with a proper choice of the hash function $h_i$. Can the authors make a more careful comparison with the "green-red" list approach along this line?
3. My understanding is that speculative sampling enhances computational efficiency, but it may not necessarily improve statistical efficiency or robustness. It would be helpful if the authors could provide intuitive explanations for how using the proposal LLM and speculative sampling improves the method's efficiency and resilience. What is the most essential component of SEAL that contributes to its statistical efficiency and robustness?
4. Given that SEAL is constructed based on maximum coupling, does it enjoy certain theoretical optimality regarding power/Type II error?
5. Please comment on the sensitivity of SEAL to the tuning parameters such as $K$, $\Omega_h$, and the hash function.

---

> ### Author Response · Authors · 2024-11-26
> **Response by Authors (1/2)**
>
> Thank you for your feedback. Please find our detailed responses below.
>
> ## Practical relevance of theoretical results
> Theorem C.2 and Theorem 2.3 focuses on theoretical study where the main target is to characterize the statistical rates of watermarking a random source. In response to your concern, **our result Theorem 2.4 deals with the practical setting where the tokens are non-iid**, which improves theoretical understanding on the number of tokens and the empirical entropy required to detect the watermark (compared with existing bounds such as Christ et al. (2023)).
>
> ## Theoretical results ignore the prompts' impacts
> All of our results hold universally for any arbitrary prompt. **The prompt influences the rates through the entropy h (or empirical entropy $\hat H$) of the text distribution**: for questions such as ‘1+1 = ?’ the entropy is low and so our rates indicate that more tokens are required to watermark responses to such prompts, and for open-ended questions the entropy could be high and the number of required tokens becomes lower according to our rates. Therefore, our rates are influenced by the prompts and our theoretical results do NOT ignore the prompts' impacts.
>
> ## The proposed SEAL method's rationale and novelty
> The inspiration comes from Theorem 2.1, where we showed that best efficiency is achieved by generating random seeds from the same watermarked model and maximally coupling the seeds with next tokens. However in practice, the watermarked model is unknown to detectors, so we adopt a proposal model which is close to all language models since they are all trained to fit the human language distribution. In this way, the random seeds become adaptive to the semantic information in preceding texts, as opposed to fixed distributions in previous works.
>
> The major novelty of SEAL is introducing speculative sampling into watermarking, which enables the random seeds to capture semantic information with provable type-I error and distortion guarantees. To our knowledge, our work is the first to use proposal LLMs and speculative sampling in watermarking LLMs.
>
> While there are existing works on semantic-aware watermarking, the major difference is that **existing semantic-aware schemes do not have formal Type I error guarantees while our watermarks have provable type I error bound**. For example, Fu et al. (2023) add some semantically-similar tokens into the green list, which makes the green list not fully (pseudo)-random; Hou et al. (2023) performs rejection sampling to sample the sentences conditional on certain green region in the semantic space, which significantly increases the probability that a human-generated token falls into the green list; Ren et al. (2023) uses a trained MLP to generate semantic values, which lacks theoretical control. In summary, we have yet seen any formal Type I error or distortion-free statements in these works, possibly due to the heuristic design. In comparison, our algorithm enjoys clear Type I error guarantee by introducing speculative sampling techniques into watermarking.
>
> ## The role of the proposal LLM in their framework and how its quality affects SEAL's performance?
> We tested the impact of the proposal LLM to SEAL's performance, summarized in the following table. We observe that the proposal model has very marginal impacts on SEAL's performance. We hypothesize that this is because in SEAL the proposal model only attends to the last K tokens, and so the fixed context window size can reduce the vibration of the next token probability among different proposal models.
> | Temperature | Proposal              | Quality (↑) | Size (↓) | Tamper-resistance (↑) |
> |-------------|-----------------------|-------------|----------|-----------------------|
> | 0.3         | TinyLlama-1.1B       | 0.89        | 63       | 1.0                   |
> |             | Phi-3-mini (3.8B)    | 0.90        | 57.5     | 1.0                   |
> |             | vicuna-7b-v1.5       | 0.89        | 61       | 1.0                   |
> | 0.7         | TinyLlama-1.1B       | 0.89        | 49       | 1.0                   |
> |             | Phi-3-mini (3.8B)    | 0.88        | 45.5     | 1.0                   |
> |             | vicuna-7b-v1.5       | 0.89        | 48       | 1.0                   |
> | 1.0         | TinyLlama-1.1B       | 0.87        | 76       | 1.0                   |
> |             | Phi-3-mini (3.8B)    | 0.88        | 75       | 1.0                   |
> |             | vicuna-7b-v1.5       | 0.87        | 72       | 1.0                   |

---

> ### Author Response · Authors · 2024-11-26
> **Response by Authors (2/2)**
>
> ## Can the authors make a more careful comparison with the "green-red" list approach along this line?
> When the proposal LLM is the target LLM, the hash function is identity mapping, and the attention window K is infinite, SEAL reduces to the statistically optimal watermark in Theorem 2.1. **Only when the proposal LLM is uniform distribution (i.e. infinite temperature) and $\Omega_h = \{0,1\}$, SEAL would reduce to the "green-red" list scheme in Kirchenbauer et al. (2023)**. In all our experiments, SEAL uses a nontrivial proposal LLM with normal temperature and outperforms the "green-red" list approach.
>
> ## What is the most essential component of SEAL that contributes to its statistical efficiency and robustness?
> SEAL enhances efficiency because its random seeds come from a proposal LLM rather than a fixed distribution. In one extreme, when the proposal LLM is the target LLM, the hash function is the identity mapping, and the attention window K is infinite, SEAL reduces to the statistically optimal watermark in Theorem 2.1. In the other extreme when the proposal LLM is uniform distribution (i.e. infinite temperature) and $\Omega_h = \{0,1\}$, SEAL would reduce to the "green-red" list scheme in Kirchenbauer et al. (2023). Therefore, **SEAL serves as an interpolation between the statistically-optimal watermarking and the "green-red" list scheme, enjoying both statistical efficiency and practicality**.
>
> SEAL enhances robustness because its random seeds depend on semantic information in preceding texts, which is more robust to paraphrasing. For example, the proposal model might generate the same next token despite the preceding tokens being paraphrased, thus resulting in the same random seed which enables successful detection.
>
> ## Given that SEAL is constructed based on maximum coupling, does it enjoy certain theoretical optimality regarding power/Type II error?
> When the proposal LLM is the target LLM, the hash function is identity mapping, and the attention window K is infinite, SEAL reduces to the statistically optimal watermark in Theorem 2.1, therefore enjoys statistical optimality. In practice, SEAL cannot achieve optimality because the target LLM and the prompt is inaccessible to the detector. However, it still improves statistical efficiency upon existing works.
>
> ## Please comment on the sensitivity of SEAL to the tuning parameters such as K, Ωh, and the hash function.
> In experiments we find that a suitable value of K lies between 10 and 20 and further increasing K will only reduce efficiency (because SEAL needs K tokens to warm-up). SEAL is observed to perform well when $\Omega_h$ is less than 5. The hash function has little influence on SEAL’s performance.

---

> > ### Comment · Reviewer_dDvN · 2024-12-03
> >
> > I thank the authors for addressing my comments and making changes to the manuscript. As a result, I have increased my score.

---

### Official Review · Reviewer_y8pq · 2024-11-04

**Soundness:** 3
**Presentation:** 3
**Contribution:** 3
**Rating:** 8
**Confidence:** 3

**Summary:**

The paper makes two main contributions: 1) it establishes (essentially) matching asymptotic upper and lower bounds on watermark statistical power (in terms of the number of tokens required to detect the watermark), where the upper bound is substantially stronger than previous work; and 2) it develops a watermark inspired by this analysis and empirically validates its power. In both cases, the main idea is to assume some knowledge of the underlying watermarked language model (i.e., the token probabilities in the putative text) and incorporate this knowledge into watermark detection.

**Strengths:**

The paper proposes a watermarking scheme and provides both theoretical and experimental evidence arguing this scheme is stronger (in terms of statistical power) than prior work. In terms of the theory, the upper bounds established in the paper are notably stronger than prior work. The experimental results are more mixed: the watermark outperforms some prior work in some settings; however, it is good that the paper at least carries out experiments to validate the theory.

**Weaknesses:**

The comparisons (both theoretical and empirical) to prior work are arguably unfair for the following reasons: 1) the paper assumes knowledge of the watermarked language model (either directly or via a proxy proposal LLM) during detection, which prior work does not; and 2) the watermark generation procedure proposed by the paper appears to be more computationally intensive than prior work (e.g., it involves using a proposal LLM and speculative decoding).

If only for clarity reasons, it would be good if the paper could be more explicit about what trade-offs exist between their method versus prior work (both theoretically and empirically). Concretely, in terms of experiments, it would be helpful to 1) systematically vary the degree to which the proposal LLM differs from the original watermarked LLM and study the effects on watermark power and 2) benchmark the runtime of watermark generation compared to prior work.

**Questions:**

Are there any guarantees regarding the difference in the *joint* distribution over multiple samples of original versus watermarked text from the watermark proposed in the paper?

---

> ### Author Response · Authors · 2024-11-26
> **Response by Authors**
>
> Thank you for your feedback. Please find our detailed responses below.
>
> > the paper assumes knowledge of the watermarked language model (either directly or via a proxy proposal LLM) during detection, which prior work does not
>
> In our theoretical results, we distinguish between two scenarios: instance-dependent and minimax. In the instance-dependent case, we derive tight rates that depend on the specific watermarked model. In the minimax case, we derive tight rates that depend on the model class to which the watermarked model belongs. Importantly, in both scenarios, **our results hold universally for any arbitrary watermarked model or model class**. Consequently, **our findings do not rely on assumptions about the models themselves but instead highlight the dependency of rates on certain complexity parameters**. Many prior works such as Aaronson (2022) and Zhao et al. (2023) also exhibit such complexity-dependency.
>
> In practical results, our algorithm is based on certain knowledge of the watermarked language model, but we think that **this is a strength rather than a weakness**. SEAL exploits the fact that language models mimic the distribution of human language instead of some arbitrary distribution. Since the proposal model and the watermarked model are both trained to fit the language distribution, they should be reasonably close. This knowledge helps us address the problem of watermarking LLMs.
>
> > what trade-offs exist between their method versus prior work (both theoretically and empirically)
>
> Theoretically, our method takes no more than 2X compute compared to standard generation because we simply add another language model for inference. Since our method couples the proposal model and the watermarked model using speculative sampling, many techniques in speculative decoding can be applied to further speed up our method, which might even end up being faster than standard generation due to parallel verification in speculative decoding. This would be an interesting future work to study.
>
> > Are there any guarantees regarding the difference in the joint distribution over multiple samples of original versus watermarked text from the watermark proposed in the paper?
>
> If the pseudo-randomness sources are considered as independent from a true random oracle, then multiple samples of watermarked texts are independent of each other and then the joint distribution is also distortion-free.
>
> Our watermarking scheme is amenable to any design of pseudo-randomness sources, and our scheme enjoys the cryptographical indistinguishability of whatever randomness sources it uses. For experiments we use text-dependent (embedded) randomness, same as many previous works such as Aaronson and Kirchner (2022) and Kirchenbauer et al. (2023). If different private keys are used, then it is computationally intractable to distinguish the joint distribution over multiple watermarked samples from the original ones.

---

> > ### Comment · Reviewer_y8pq · 2024-11-26
> >
> > The authors have adequately addressed most of my questions and I have adjusted my score accordingly.

---

> > > ### Author Response · Authors · 2024-11-27
> > >
> > > Thank you very much for your active engagement in discussion and thoughtful feedback!

---

### Official Review · Reviewer_XWGd · 2024-11-07

**Soundness:** 2
**Presentation:** 2
**Contribution:** 3
**Rating:** 5
**Confidence:** 4

**Summary:**

The paper contains two independent contributions:

The theoretical analysis provides a treatment of watermark strength in terms of Type I and II errors, discusses the best watermark in the minimax sense, i.e., how much Type II error can be achieved while ensuring Type I error is bounded, and the optimal watermark detection method to achieve it, both when the model distribution is known and when it is unknown but has a bounded probability (so the probability is sufficiently non-concentrated). It then explores the minimum number of tokens needed to achieve bounded Type I and II errors in both iid and non-iid settings.

The practical SEAL method introduces an additional proposal model and assumes its output distribution is known. It uses maximal coupling to couple the proposal and target models during generation, similar to speculative sampling, and uses the proposal model's output distribution for detection.

**Strengths:**

Provides a theoretical treatment of watermark strength in terms of Type I and II errors.

Derives minimax optimal watermarks and detection methods under different knowledge assumptions about the model distribution.

Gives bounds on the number of tokens needed for effective watermarking in various settings.

Proposes the SEAL method that leverages a proposal model.

**Weaknesses:**

1. Insufficient comparison with related semantic-aware watermarking methods:

The paper cites several semantic-aware watermarking methods, such as:
-  "Watermarking Conditional Text Generation for AI Detection: Unveiling Challenges and a Semantic-Aware Watermark Remedy", which uses a word similarity matrix
- "SEMSTAMP: A Semantic Watermark with Paraphrastic Robustness for Text Generation" and "k-SemStamp: A Clustering-Based Semantic Watermark for Detection of Machine-Generated Text", which use locality-sensitive hashing
- "A robust semantics-based watermark for large language model against paraphrasing", which uses Normalized Embedding Ring (NE-Ring)

All these methods seem to better capture semantic information compared to the random hash functions used in this paper. However, the paper does not provide any discussion or comparison with these cited works. The random hash functions and other designs in SEAL are hard to justify as capturing semantic information. More details in point 6 below.

2. Lack of rigor in theoretical proofs:

- In the proof of Theorem C.2, in line 1435, authors claim "third inequality is achieved when $\sum_{x\in\Omega:\rho(x)<\alpha}\left(\alpha-\rho(x)\right)\geq\epsilon$. a sufficient condition for which being $|\Omega|\geq1/\alpha $".

However, this is clearly not a sufficient condition as it is missing an ϵ term.

- In Theorem C.14, in line 1888, the proof uses a new condition $m:=|\Omega|\gtrsim1/\kappa$ or equivalently $n\log|\Omega_0|\gtrsim\widehat{H}$ which was not given in the theorem statement.

- Again, in Theorem C.14, line 1893, it applies the result of Theorem C.6 without ensuring its conditions are met.

- Again, in Theorem C.14, line 1912, the proof is omitted with the authors claiming "some arithmetic shows". However, the steps are hard to establish. It is unclear how to verify the first inequality
$\max\\{\frac{\binom{m-\frac{\rho(\bar{\Omega})}{\kappa}}{\alpha m}}{\binom{m}{\alpha m}}-\rho(\bar{\Omega}^{c}),0\\}\leq\rho(\bar{\Omega})\cdot(1-\alpha)^{1/\kappa\_{0}}$
 without further explanation from the authors. The second inequality $\rho(\bar{\Omega})\cdot(1-\alpha)^{1/\kappa_0}\leq\beta\cdot\rho(\bar{\Omega})$ where $\kappa_{0}=\operatorname{log}\frac{1}{\alpha}+\operatorname{log}\operatorname{log}\frac{1}{\beta}$ has clear counterexamples, e.g., $\alpha=e^{-1}, \beta=e^{-e}, \kappa_0=2$ gives $(1-\alpha)^{1/\kappa_0}\approx 0.794$ and $\beta\approx 0.06599$. It is completely unclear how the proof proceeds.

- The Type I error bound for SEAL relies on a quantile of sum of Bernoulli random variables that are assumed to be independent, as stated in line 1242. However, this is impossible as the mean of $\xi_{i+1}$ at the $(i+1)$-th token depends on the proposal model's output distribution, hash function, and ultimately the prefix $t_{1:i}$, which is correlated with $\xi_i$. So they are not i.i.d. and this assumption cannot be used. While modeling as i.i.d. Bernoulli may be a convenient assumption that gives practical watermark detection criterion, it is not theoretically rigorous and should not be presented as a strict theorem but rather an educated guess.

- Additionally, Theorem G.5 is not rigorously stated. The assumption "When the sequence y is not generated by Algorithm 3, ξj 's are i.i.d. Bernoulli random variables" is too weak, as "not generated by Algorithm 3" itself does not imply any information. It is entirely possible to generate from a slightly perturbed version of Algorithm 3, in which case the theorem cannot possibly hold.

3. Outdated and lacking experimental baselines:

   - Although the paper claims to "compare SEAL with one of the state-of-the-art watermarking methods, the exponential scheme (Aaronson, 2022b)", Aaronson's work is a pioneer work and definitely not the latest SOTA. The claim of it being SOTA is incorrect.

   - The paper completely lacks comparison with the semantic-aware methods listed above. These methods are relevant to SEAL as they all deal with the semantic-aware direction, which SEAL claims to do.

   - The paper also lacks comparison with model non-agnostic methods. Table 5 is made under the model non-agnostic setting, but all the compared works are model agnostic methods that do not actively utilize model probability information, which is misleading and unfair. It should have included existing model non-agnostic methods, such as:
     - The log likelihood ratio (LLR) test, which is the strongest test as discussed in "Unbiased Watermark for Large Language Models" and "A statistical framework of watermarks for large language models: Pivot, detection efficiency and optimal rules."
     - The Maximin variant of LLR score from "Unbiased Watermark for Large Language Models", which is a robust practical variant of LLR.

   - The lack of thorough review/comparison leads to an incorrect claim in line 97 that "previous rates on the number of tokens consistently fails to surpass $h^{-2}$", as the authors completely overlooked model non-agnostic methods (e.g. LLR) that already achieve the best detection efficiency.

4. Missing key experimental details:

   - The experiments lack error bars, which are important for readers to check what is signal and what is noise. It is hard to do so without reporting error bars.

   - When comparing exponential and SEAL, both are unbiased (distortion-free) in token distribution but the plotted quality differs a lot. If there are no implementation errors, this suggests a large variance that the authors should have clearly plotted. The errors are also missing in Table 1.

   - The experiments are suspicious. In Table 1, Exponential, Inverse Transform, Binary, and SEAL can all achieve unbiased (distortion-free) token distribution, but the quality differs drastically. It is reasonable to suspect issues with the experiments, but the authors do not provide code and there is no way to confirm.

5. Weak connection between the theoretical and algorithmic contributions:

   - The authors claim "This insight led us to develop semantic-aware random seeds, which adapt to the underlying distribution of the tokens." (line 264). However, Section 2 and the appendix do not mention any discussion about semantics, and the word "semantic" is not even mentioned.

   - The authors also claim "Second, our analysis of both Theorem 2.3 and 2.4 reveals ... and the pushforward of the token distribution onto the same measure space". This is not obvious from Theorem 2.3 and 2.4, which do not even mention the word "pushforward". Further explanation is needed on how the theory in Section 2 provides information about the pushforward measure.

   - There are many other mismatches:
     - Section 2 has a lot of discussion on Type II error, but Section 3 does not discuss Type II error at all.
     - The final result in Section 2 is deriving the optimal rate $h^{-1}\cdot\log(1/h)$, but Section 3 does not discuss whether this rate can be achieved. Reading the abstract, one would expect a new algorithm achieving the optimal rate, which would be an important progress, but there is no relevant discussion in Section 3.

   - It is unclear why Section 2 is related to Section 3. It seems like two independent contributions stitched together.

6. SEAL does not truly capture semantic information:

   - SEAL still operates on token distributions like traditional watermarks (but with a projection onto hash codes space with a random hash function). The hash functions, speculative sampling (maximal coupling) in this process do not handle semantic information such as distinguishing word similarities. Only the original language model and proposal model use semantic information when doing next word prediction. A watermarking method should not be called semantic-aware just because next word prediction $p(\cdot|x_{1:t})$ contains semantic information.

   - Examples of truly semantic-aware watermarking methods:
     - "SEMSTAMP: A Semantic Watermark with Paraphrastic Robustness for Text Generation" and "k-SemStamp: A Clustering-Based Semantic Watermark for Detection of Machine-Generated Text" use hash functions that consider semantic information.
     - "Watermarking Conditional Text Generation for AI Detection: Unveiling Challenges and a Semantic-Aware Watermark Remedy" combines semantically similar words when watermarking.
     - "A robust semantics-based watermark for large language model against paraphrasing" extracts watermark information from embeddings.

     All of these methods better capture semantic information compared to the random hash functions used in this paper.

   - In contrast, the random hash functions and speculative sampling in SEAL do not handle semantic information and should not be called "Semantic aware".

7. Writing issues:

   - Inconsistency between algorithm and figure: In the "Bootstrapping efficiency with logits bias" part, it is stated that a bias δ is introduced to the logits, but Figure 2 shows it being added in the maximal coupling step.

   - Typos and undefined notations:
     - In line 812, G is not defined.
     - In line 825, C is defined but not used. It does not appear in the actual steps of the algorithm or the definition of the rejection region R. It is unclear why it is defined and what its purpose is.

   - Confusing notation: $\eta$ is used to denote minimax in the main text (line 185, line 206) but used to denote probability in the appendix (line 785).

**Questions:**

1. Can the theorems and proofs be fixed?

2. How does SEAL's random hashing compare to prior locality-sensitive hashing, word similarity matrix, or Normalized Embedding Ring methods in capturing semantic information? Why is next word token prediction following random hash functions considered to capture semantic information?

3. Why is SEAL considered model-agnostic when it requires a language model's probabilities? If a watermarking algorithm generates watermarks using model M1 and detects watermarks using model M2, is it still considered model-agnostic?

4. Is there any deep connection between the Type II error theoretical analysis and SEAL?
   - Half of the main text discusses the asymptotic number of required tokens, with the optimal rate being $h^{-1}\cdot\log(1/h)$. Can SEAL achieve this rate proposed in the paper? If so, why is there no theoretical proof?
   - Or is it the same as all existing model-agnostic methods (like the exponential method), i.e., $h^{-2}$?
   - The paper claims that the theoretical analysis guided the design of the new SEAL method. Does SEAL achieve any theoretical asymptotic bound improvements through this guidance?

5. With what probability is the hash function introduced in the SEAL algorithm not injective?
   - If the hash codes space $\Omega_{h}$ is very large, is a random hash function injective with high probability?
   - If it is injective, is the entire process equivalent to normal speculative sampling?
   - If the hash function step is skipped and speculative sampling is performed directly on token space, will the watermark strength be stronger?
   - Why introduce this extra hash step?

---

> ### Author Response · Authors · 2024-11-26
> **Response by Authors (1/2)**
>
> Thank you for your feedback. Please find our responses below.
>
> ## Comparison with related semantic-aware watermarking methods
> We cited some semantic-aware watermarking methods and discussed our differences in Section 1.1. While there are existing works on semantic-aware watermarking, the major difference is that existing semantic-aware schemes do not have formal Type I error guarantees while our watermarks have provable type I error bound. For example, Fu et al. (2023) add some semantically-similar tokens into the green list, which makes the green list not fully (pseudo)-random; Hou et al. (2023) performs rejection sampling to sample the sentences conditional on certain green region in the semantic space, which significantly increases the probability that a human-generated token falls into the green list; Ren et al. (2023) uses a trained MLP to generate semantic values, which lacks theoretical control. In summary, we have yet to see any formal Type I error statements in these works, possibly due to the heuristic design. In comparison, our algorithm enjoys clear Type I error guarantee by introducing speculative sampling techniques into watermarking.
>
> ## Proofs of theorems
> In Theorem C.2, $\geq$ should be $\gtrsim$ and the sufficient condition more precisely is $|\Omega| \geq (2+\epsilon)/\alpha$. Indeed, if the desired inequality is not satisfied then $1 \geq \sum_{x:\rho(x) < \alpha} \rho(x) >  \alpha \cdot (|\Omega| - 1/\alpha) - \epsilon \geq 1$, a contradiction.
>
> In Theorem C.14, $\kappa_0$ should be defined as $\kappa_0 = \alpha/\log \frac{1}{\beta}$, and what we need are $\kappa := e^{-\widehat H} \lesssim \kappa_0$ and $m := |\Omega| = |\Omega_0|^n \gtrsim 1/\kappa_0$. Line 1893 is guaranteed by the definition of $\eta^*$ and does not apply the result of Theorem C.6. Line 1912 holds with the corrected definition of $\kappa_0 = \alpha/\log \frac{1}{\beta}$. In the revision, we have fixed the typos and added necessary explanations to each step.
>
> In line 1242, the Bernoulli random variables are indeed independent because we are at the detection phase under H_0. We understand your point that at the generation phase $\xi_j$’s are generated autoregressively and should be defined on filtrations. However, at the detection phase under H_0, the token sequence $y$ is fixed and the randomness comes solely from the proposal model’s output. As shown in Algorithm 4 line 5, the distributions $\rho_j$’s are independent of each other, and so are $\xi_j$’s.
>
> In Theorem G.5, we change the statement to ‘For any fixed (sub-)sequence $y$’, which matches more precisely with our definition of Type I error in Problem B.1.
>
> ## Log likelihood ratio (LLR) test and other model non-agnostic methods
> The UMP watermark in Table 5 is equivalent to the Log likelihood ratio test. As clarified in lines 1389-1391, Table 5 only serves to exhibit optimal rates rather than comparison. We have made these points clearer in revision.
>
> ## Experimental details
> We conducted more experiments with varying secret keys. The error bars of SEAL are shown in the following table:
> | Temperature | Quality (↑)          | Size (↓)       | Tamper-resistance (↑) |
> |-------------|-----------------------|----------------|-----------------------|
> | 0.3         | 0.893 ± 0.004   | 57.5 ± 2.5     | 1.0 ± 0              |
> | 0.7         | 0.88 ± 0.003         | 45.3 ± 1.25    | 1.0 ± 0              |
> | 1.0          | 0.873 ± 0.007        | 73.3 ± 1.5     | 1.0 ± 0              |
>
> The numbers of baseline schemes are retrieved directly from the benchmark of Piet et al. (2023). We conducted experiments with the provided parameters and found that the quality of baseline schemes is around 0.89, closer to SEAL. We hypothesized that it is due to Piet et al. (2023) performing some hyperparameter search to optimize the quality metric.
>
> In experiments, we use the biased version of SEAL with nonzero $\delta$. The hyperparameter details can be found in Appendix E. Therefore, it is reasonable that SEAL exhibits certain quality degradation.
>
> ## Connection between the theoretical and algorithmic contributions
> The pushforward of the token distribution onto the same measure space is $\eta^*\left(p\left(W \cap (U \times 2^\Omega)\right)\right)$ in Eq.(13).
>
> SEAL is inspired from Theorem 2.1, where we showed that the best efficiency is achieved by generating random seeds from the same watermarked model and maximally coupling the seeds with the next tokens. However, in practice, the watermarked model is unknown to detectors, so we adopt a proposal model that is close to all language models since they are all trained to fit the human language distribution. In this way, the random seeds become adaptive to the semantic information in preceding texts, as opposed to fixed distributions in previous works. The random seed is the hashing of the proposal model’s next-token prediction. To couple the random seed and the true next token, we use ideas from Theorem C.6.

---

> ### Author Response · Authors · 2024-11-26
> **Response by Authors (2/2)**
>
> ## Capture semantic information
> We call a watermarking method semantic-aware if its random seed’s distribution is dependent on the previous texts. In SEAL, the random seed is the hashing of the proposal model’s next-token prediction, and so depends on previous texts. In comparison, Kirchenbauer et al. (2023)’s random seeds (green lists) are uniformly random subsets of the vocabulary, Aaronson (2022b)’s random seeds follow uniform distribution, Christ et al. (2023)’s random seeds are Bernoulli random variables, all following fixed distributions. Therefore, we say that SEAL is semantic-aware as opposed to baseline schemes.
>
> ## Why is SEAL considered model-agnostic
> According to Problem B.4, SEAL is model-agnostic because the random seed’s distribution (derived from the proposal model) is independent of the particular watermarked model. SEAL uses the same proposal model in generation and detection: if one use M1 as proposal model in generation and M2 in detection then they would likely not detect any watermark.
>
> ## Type II error theoretical analysis of SEAL
> When the proposal LLM is the target LLM, the hash function is identity mapping, and the attention window K is infinite, SEAL reduces to the statistically optimal watermark in Theorem 2.1, therefore enjoys statistical optimality. In practice, SEAL cannot achieve optimality because the target LLM and the prompt is inaccessible to the detector. However, it still improves statistical efficiency upon existing works.
>
> ## Hash function
> As discussed above, SEAL’s semantic awareness is not attributed to the random hashing function.
>
> Similar to Kirchenbaur et al. (2023), we select a small hash code space in SEAL. This means that the hash function is not injective.
> If the hash function is injective, the generation process is equivalent to speculative sampling. But we empirically found that the performance wouldn’t be stronger with injective hash function.
>
> The hash function increases the probability of hash collision and thus enhances robustness. We empirically found that the SEAL works best when the hash code space size is small (<= 5).

---

> > ### Comment · Reviewer_XWGd · 2024-11-27
> >
> > First I want to thank authors for the response. However, many points in my review are not fully addressed.
> >
> > > Insufficient comparison with related semantic-aware watermarking methods:
> >
> > The authors' response does not adequately address the concern about the lack of discussion and comparison with the cited semantic-aware watermarking methods. The response focuses on the difference in formal Type I error guarantees but does not provide a direct comparison of the semantic-awareness aspect, which was the main concern raised in my review.
> >
> > > Lack of rigor in theoretical proofs
> >
> > First I want to thank authors for comment in the response on how to fix the proof. However, when I check the first point of Theorem C.2, I found the revised paper still have the old proof without the fix.
> >
> > > Outdated and lacking experimental baselines
> >
> > The authors' response does not address the concerns about the lack of comparison with semantic-aware methods and model non-agnostic methods. The response only clarifies that the UMP watermark in Table 5 is equivalent to the Log likelihood ratio test (which I cannot find from lines 1389-1391 as indicated by authors), but it does not discuss other model non-agnostic methods.
> >
> > > lack error bars
> >
> > The authors have provided error bars for SEAL in their response, addressing one of the concerns raised in my review. However, the response does not adequately address the issue of the large difference in quality between unbiased methods, despite the small error bars in the additional experiments:
> >
> > "When comparing exponential and SEAL, both are unbiased (distortion-free) in token distribution but the plotted quality differs a lot. If there are no implementation errors, this suggests a large variance that the authors should have clearly plotted."
> >
> > > Why is SEAL considered model-agnostic
> >
> > If I understand correctly, SEAL needs to use a language model, which is expected to be similar to the target model to have good watermarking property. I wouldn't call SEAL model-agnostic, because it indeed uses a language model.
> >
> > > Writing issues
> >
> > The authors' response and paper revision does not address the writing issues mentioned in the review.

---

> ### Author Response · Authors · 2024-11-27
>
> Thank you for engaging in the discussion and providing your thoughtful feedback. We appreciate your detailed comments, which have helped us further refine our manuscript. Below, we address each of your points:
>
> > Comparison with related semantic-aware watermarking methods
>
> We have clarified that:
> (i) Related semantic-aware watermarking methods are already discussed in Section 1.1.
> (ii) The previous semantic-aware schemes **belong to a different domain as they lack formal Type I error guarantees**.
> It is important to emphasize that **our work specifically focuses on statistical watermarking with formal guarantees**, which distinguishes it from existing semantic-aware methods. These points have been further elaborated in the revised manuscript.
>
> > Experimental baselines
>
> As noted, **existing semantic-aware methods fall outside the scope of our study**. Our focus is on statistical watermarking methods with formal guarantees, and while there are numerous watermarking approaches (q.v. the review of Tang et al. (2023)), it is infeasible to compare against all of them.
>
> We also reiterate that the UMP test described in Theorem 2.1 is essentially a likelihood ratio (LLR) test, and we are unaware of other model-nonagnostic methods (Hu et al. (2023)'s watermark is detectable in a likelihood-agnostic way, which makes it not model-nonagnostic). Furthermore, as noted below Table 5, the results in Table 5 are intended to demonstrate statistical limits in Type I/II errors, rather than serve as comparison or advocate for the superiority of a particular watermarking approach.
>
> > Difference in quality between unbiased methods
>
> As we clarified in responses, the numbers of baselines in table 1 come directly from the benchmark report (which according to their  report are tuned specifically for quality metric) and our own further experiments demonstrate that the quality of baseline schemes is approximately 0.89, close to SEAL. This suggests **there is no significant quality difference between SEAL and baseline methods**.
>
> Additionally, we have clarified that we employ a biased version of SEAL with nonzero $\delta$. Therefore, it is incorrect to say that "When comparing exponential and SEAL, both are unbiased (distortion-free) in token distribution but the plotted quality differs a lot. If there are no implementation errors, this suggests a large variance that the authors should have clearly plotted."
>
> > Why SEAL is considered model-agnostic
>
> As noted in Problem B.4, a watermarking scheme is considered model-agnostic if the distribution of the random seed (derived from the proposal model) is independent of the particular watermarked model. SEAL adheres to this definition and is therefore classified as model-agnostic.
>
> **Our definition aligns with Kuditipudi et al. (2023)'s original definition of model-agnosticity, where they defined as 'agnostic—it should be detectable without the language model and/or prompt'**. The term "*the* language model" in their definition refers specifically to the watermarked model, not to any arbitrary language model. Consequently, a method is model-agnostic even if it utilizes another language model for watermark generation or detection.
>
> SEAL's approach leverages another language model for this purpose because all language models are trained on human language distributions, which ensures reasonable similarity. This **stems from the nature of the problem rather than an additional assumption: we aim to watermark outputs of language models, not arbitrary sequences**.
>
> > Writing issues and theorem proofs
>
> Thank you for your feedback. In the revision, we have changed the notation $\eta$ in the main body to avoid conflicts and we have updated the proof of Theorem C.2.
>
> Adding a bias δ to the logits is in the maximal coupling step. This is clear as the **"Bootstrapping efficiency with logits bias" paragraph is under Section 3.1.2: Maximal Coupling Construction**. Thus, there is no inconsistency between the algorithm and figure.
>
> In line 812, $L_G$ has been defined in the step above: "Randomly partition $V$ into a green list $L_G$ of size $\gamma |V|$, and a red list $L_R$ of size $(1 - \gamma)|V|$."
>
> Line 825: $C$ is the threshold for the z-test, as referenced in Eq. (2) of Kirchenbaur et al. (2023). We have clarified this parameter in the revision.
>
> Thank you again for your constructive input. We hope these responses address your concerns and look forward to your further insights.

---

### Author Response · Authors · 2024-11-28
**Summary of Revision**

We sincerely thank all reviewers for their thoughtful and constructive feedback, which has significantly improved the quality of our work. In response to the comments, we have made the following major revisions:

- *Expanded Discussion on Semantic-Aware Methods*: In Section 1.1, we have incorporated additional discussions on existing semantic-aware watermarking schemes, clarifying how our work differs from these approaches in scope and focus.

- *Additional Experiments on SEAL and Baseline Schemes*: We conducted further experiments on SEAL and baseline methods under different private keys. These experiments allowed us to compute error bars for the metrics, refine the configuration of SEAL, and replace previously reported values (retrieved from Piet et al. (2023)) in Figure 1 and Tables 1 & 2.

- *Ablation Study on Proposal Models*: In Section F.1, we included an ablation study to assess the impact of proposal model size, exploring models ranging from 1B to 7B parameters.

- *Enhanced Explanation of SEAL*: We have further elaborated on the connection between theory and SEAL throughout the main body, and provide clearer explanations of its novelty, mechanism, and comparisons (both theoretical and empirical) to prior works.

- *Improvements to Writing and Proofs*: We revised the proofs of Theorems C.2, C.14, and G.5. We also improved the overall writing of the paper, resolving ambiguities raised by reviewers.

We deeply appreciate the reviewers’ invaluable input and believe these revisions address their concerns comprehensively.

---

### Meta-Review · Area_Chair_uQ6W · 2024-12-19

**Metareview:**

Summary: This paper studies the problem of implanting watermarks into LLM outputs. The paper analyzes the Type I and Type II errors, minimizing the Type II error while requiring the Type I error to be below a threshold. A new watermarking method, SEAL, is proposed based on semantic-aware speculative sampling.

Strengths:
1. The paper provides a minimax theoretical analysis for the optimality of watermarks.
2. The paper proposes a practical method SEAL for LLM watermarking, with experimental verification of its advantages.

Weaknesses:
1. Reviewers have concerns on insufficient and unfair comparison with related semantic-aware watermarking methods (as raised by Reviewer y8pq and Reviewer XWGd).
2. Reviewers believe the proofs lack of rigor, and the revised version fails to fix the problem.
3. Some writing issues in the paper (as raised by both Reviewer XWGd and Reviewer xR9M).

**Additional Comments On Reviewer Discussion:**

The paper is on the borderline. One reviewer is willing to champion the paper with a score of 8, while other three reviewers rate the paper with a borderline scores (two marginally below + one marginally above the acceptance threshold). AC believes Reviewer XWGd's opinion is professional, with a high-quality review and response after rebuttal. After reading the rebuttal, Reviewer XWGd keeps his/her original score of 5, as his/her following concerns remain:

1. insufficient comparison with related semantic-aware watermarking methods.
2. The proofs lack of rigor, and the revised version fails to fix the problem with an old proof.
3. The rebuttal does not adequately address the issue of the large difference in quality between unbiased methods, despite the small error bars in the additional experiments.
4. Some writing issues remain.

The strengths in Reviewer y8pq's review are standard and do not stand out according to AC's judgement. Given two negative votes out of four reviews and Reviewer XWGd's high-quality review, AC would recommend rejection (however, with low confidence). SAC agreed with the AC for this decision.

---

### Decision · Program_Chairs · 2025-01-22

Reject